Colorimetric derivatization of ambient ammonia ($NH_3$) for detection by
long path absorption photometry
Shasha Tian[a, b, 1], Kexin Zu[a, b, 1], Huabin Dong[a, b, *], Limin Zeng [a, b], Keding Lu [a, b], Qi Chen [a]
[a] State Key Joint Laboratory of Environmental Simulation and Pollution Control, College of
Environmental Sciences and Engineering, Peking University, Beijing, 100871, China.
[b] International Joint laboratory for Regional pollution Control (IJRC), Peking University, Beijing, China
* Corresponding author: hbdong@pku.edu.cn
**Abstract.** In the last few decades, various techniques, including spectroscopic, mass spectrometric,
chemiluminescence, and wet chemical methods, had been developed and applied for the detection of
gaseous ammonia ($NH_3$). We developed an online $NH_3$ monitoring system—salicylic acid derivatization
reaction and long path absorption photometer (SAC-LOPAP)—based on a selective colorimetric reaction
to form a highly absorbing reaction product and a LOPAP, which could run stably for a long time and be
applied to the continuous online measurement of low concentrations of ambient $NH_3$ by optimizing the
reaction conditions, adding a constant temperature module and liquid flow controller. The detection limit
reached with this instrument was 40.5 ppt with a stripping liquid flow rate of 0.49 ml $min^{-1}$ and a gas
sample flow rate of 0.70 L $min^{-1}$. An inter-comparison of our system with a commercial instrument
Picarro G2103 analyzer (Picarro, US) in Beijing was presented, and the results showed that the two
instruments had a good correlation with a slope of 1.00 and an $R^2$ of 0.96, indicating that the SAC-
LOPAP involved in this study could be used for the accurate measurement of $NH_3$.

## 1. Introduction

Gaseous ammonia ($NH_3$) widely exists in the atmosphere and plays an important role in many
atmospheric chemical reactions (Swati and Hait, 2018; Klimczyk et al., 2021; Wang et al., 2018). As the
most abundant alkaline gas in the atmosphere, $NH_3$ easily forms ammonium ions ($NH_4^+$) with water and
reacts with acidic species to form secondary inorganic particles (Ianniello et al., 2011; Ni et al., 2000).
These secondary particles are considered a major source of fine particulate matter (PM), which is harmful
to climate, visibility and human health (Bu et al., 2021; Gao et al., 2021) . Furthermore, recent studies
have shown that $NH_3$ is necessary to control fine particulate pollution (Wen et al., 2018; Wang et al.,
2013). Due to those problems, the inventory of $NH_3$ emissions and concentration in urban air has been
highly evaluated. Agriculture, including animal feedlot operations, is considered as the largest emission
source of $NH_3$ with 80.6% of the global anthropogenic emissions followed by 11% from biomass burning
and 8.3% from the energy sector, including industries and traffic (Behera et al., 2013). Expert estimate
that global annual emissions of $NH_3$ will increase from 65 Tg N $yr^{-1}$ in 2008 to 135 Tg N $yr^{-1}$ in 2100
(Fowler et al., 2015). However, ambient measurement of $NH_3$ concentrations is difficult due to several
factors: ambient levels vary widely with from 5 pptv to 500 ppbv (Janson et al., 2010; Krupa, 2003;
Sutton et al., 1995). Ammonia exists in gaseous, particulate and liquid phases, which add further
complicates the measurement (Warneck, 1988). In addition, $NH_3$ is "sticky" and interacts with surfaces
of materials, resulting in slow inlet response times (Yokelson et al., 2003). Finally, the temperature
difference between the indoor and outdoor environments and the humidity difference between the inside
and outside of the instrument will reduce the accuracy of measurement and calibration. It is therefore
essential to accurately measure ambient $NH_3$ to better quantify concentration and concentration changes
and hence to evaluate the impacts of $NH_3$.

43        In recent years, researchers have developed techniques and methods for detecting $NH_3$ in the

atmosphere, which include spectroscopic, mass spectrometric, chemiluminescence, and wet chemical
methods (Von et al., 2009). Spectroscopic methods, such as Cavity Enhanced Absorption Spectroscopy
(CEAS) (Gong et al., 2017; Berden et al., 2000) and Cavity Ring-Down Spectroscopy (CRDS) (Martin
et al., 2016; Qu et al., 2012), can greatly improve spectral absorption's effective optical path length by
using the optical cavity structure. However, the "sticky" of $NH_3$ will affect background, detection
efficiency and detection response time of the instrument (Whitehead et al., 2008; Yokelson et al., 2003).
Utilizing a quantum cascade laser (QCL) or a DFB laser in a near-infrared band as the light source can
achieve a low detection limit of 0.018ppb (Whitehead et al., 2008; Mcmanus et al., 2002; Von et al.,
2009), realizing the measurement of low concentrations of $NH_3$ in ambient air. Mass-spectrography
analyzers provide highly sensitive techniques but may be less specific and can be affected by competing
ion chemistries. The chemical ionization mass spectrometer (CIMS) technique is based on an ion-
molecule reaction to selectively ionize and detect trace $NH_3$ in the atmosphere, which features a fast
response and in situ measurement (Benson et al., 2010; Nowak et al., 2007; Yu and Lee, 2012). It has
the advantages of small volume and wide measurement range, but its detection limit is very high (Ajay

and Beniwal., 2019). Chemiluminescence is an indirect method to measure ammonia. Two catalytic

converters of different characteristics catalyze NOx and NO-amine into NO. The $NH_3$ mixing ratio is

calculated by the difference between NOx and NO-amine. This method can realize the simultaneous

measurement of $NH_3$, NO and $NO_2$, but the measurement results are affected by the conversion efficiency

(Sharma et al., 2010; Sharma et al., 2012). Wet chemistry methods convert gas-phase $NH_3$ to aqueous

$NH_3$ ($NH_4^+$) for online analysis by means of online ion chromatography with a detection limit of 0.05

$\mu g\ m^{-3}$ (0.72ppb at 25 °C) (Khlystov et al., 1995; Dong et al., 2012; Makkonen et al., 2012). A field inter-

comparison of $NH_3$ measurement techniques found that wet chemistry instruments showed better long-

term stability and agreement than other analyzers (Von et al., 2009), which was due to the wet chemical

trapping method and standard calibration solutions, humidity did not affect the measurement, and the

standard solution was more stable than standard gases. However, they failed to capture the peak because

of lower time resolution. Based on a selective colorimetric reaction to form a highly absorbing reaction

product and absorption spectrophotometry collect $NH_3$ (and ammonium) by aqueous scrubbing in glass

frit impactors (Bianchi et al., 2012; Bae et al., 2007) has been used for decades for routine derivatization

and colorimetric analysis of $NH_4^+$ in a wide variety of environmental samples (e.g. soils, environmental

waters, etc), which has also been reported by other scholars (Bae et al., 2007). In those studies the product

was detected by a long path absorption photometer (LOPAP), in which the absorbance of the solution is

amplified in the long path module to reach a lower detection limit (Heland et al., 2001).

In this study, we provide an online $NH_3$ monitoring system based on wet chemistry stripping of

atmospheric $NH_3$, followed by the formation of a highly light-absorbing indophenol after a salicylic acid

derivatization reaction to produce the colored reaction product reaction and detected with LOPAP.

According to Lambert-Beer's law, the sensitivity of spectrophotometry can be enhanced by increasing

the optical path length. This sensitive analytical method has already been successfully applied in different

colorimetric detection studies (Yao et al., 1998; Heland et al., 2001; Callahan et al., 2002). In analogy to

the original long path absorption photometer (LOPAP) which was developed for HONO measurements

(Kleffmann et al., 2002), we call this monitoring system the salicylic acid derivatization reaction and

long path absorption photometer (SAC-LOPAP), which features several improvements over versions

previously reported by other groups: one is the optimization of reaction conditions, the other

modification is the use of constant temperature module and flow control system. Secondly, we will

present measurements demonstrating our new system in urban environments in Peking University, with
measure low concentrations, good stability and low detection limit.

## 2. SAC-LOPAP instrument

## 2.1 Measurement principle

Our instrument is designed to measure $NH_3$ in a low-concentration environment (under 20ppb) with the
good stability, low detection limit (less than 60 ppt) and small size. There is a brief introduction to the
principle of the instrument. The measurement of $NH_3$ in the SAC-LOPAP instrument is achieved by the
selective colorimetric reaction to form a highly absorbing reaction product and absorption
spectrophotometry. Samples containing dissolved ammonia and ammonium react with a phenolic
compound and a chlorine-donating reagent to form indophenol blue during the reaction, with the
strongest absorption at a wavelength of 665 nm. (Krom and Michael, 1980; Searle and Phillip, 1984).The
reaction mechanism of the chromogenic reactions as shown in (1)-(4). Furthermore, to measure the
absorbance of the sample, we used a LOPAP based on liquid-waveguide capillary cell (LWCC)
technology to obtain a better detection limit, continuity and stability (Heland et al., 2001).

$$NH_3 + HOCl \rightleftharpoons NH_2Cl + H_2O \tag{1}$$

(2)

(3)

$$\text{(structure: cyclohexadienone with COO}^- \text{ and NH}_2) \quad + \quad \text{(salicylate: OH, COO}^-) \quad \xrightarrow[\text{[Fe(CN)}_5\text{NO]}^{2-}]{\text{Catalytic}} \quad \text{HO–(ring)–N=(quinone ring with COO}^-\text{, O)} \qquad (4)$$

## 2.2 Experiment setup

We designed our system to consist of four modules: the sampling module, the reacting module, the detecting module, and the control module (Fig. 1). The key component of the sampling module is a glass coil reactor, which is an open glass tube (inner diameter 1.5 mm, 75 cm long) coiled 12 turns. At the beginning of this coil, there is a flow manifold to mix the ambient air flow and the stripping solutions. The air is pumped into the stripping coil under the action of a vacuum diaphragm air pump and a gas flow meter (Horiba, China) (Chen et al., 2004). To protect the gas flowmeter and the air pump, a security bottle is installed in front of the gas flowmeter to prevent the inflow of liquid. At the same time, the stripping solution, regulated by the liquid flow control system, is injected into the stripping coil to capture $NH_3$ in the air and form a mixture of ammonium-salicylic acid. To achieve higher absorption efficiency, circulating cooling water with a temperature of 10-15 ℃ is provided outside the stripping coil. The center part of the reacting module is a reaction coil and a debubble. The liquid sample is mixed with the alkaline derivatization solution, and a derivatization reaction to produce the colored reaction product reaction occurs in the heated reaction coil. The reaction coil is made of a 90 cm length of Teflon tubes coiled on a heat-conducting metal cylinder, and a PID controller controls the temperature of the reactor at 40-75 ℃ to accelerate the derivatization reaction. After the derivatization reaction, the sample is sent to the detecting module, which comprises a liquid waveguide capillary cell (LWCC-100, World Precision Instruments, USA) with optical path length of 100 cm, an LED light source with the mode at 665 nm (Ocean Optics) and a phototube (S16008-33, HAMAMTSU, Japan) for the long path photometry detection. The sample solution to be tested is filtered by a 1.0 μm filter before passing through LWCC to avoid interference from components of the sample matrix/method reagents. Both the fluid propulsion module and detection module can be computer controlled.

Eq. (5) can help convert the concentration of $NH_4^+$ solution $C_{NH_4^+}$ to the $NH_3$ concentration in the gasous $C_{NH_3}$.

$$C_{NH_3} = \frac{C_{NH_4^+} F_l RT}{M_{NH_4^+} F_g P \gamma}$$ (5)
Where $C_{NH_3}$ denotes the content of NH₃ in the air sample (ppb), $P$ denotes atmospheric pressure
(101.3 kPa), $M_{NH_4^+}$ denotes the molar mass of NH₃ (18.04 g/mol), $R$=8.314 Pa·m³·mol⁻¹·K⁻¹. $T$ denotes
the room temperature (K), $F_l$ denotes the flow rate of stripping solution, $F_g$ denotes the flow rate of
sampling gas, $\gamma$ denotes the capture efficiency of air NH₃ in the stripping solution (a constant determined
by laboratory).

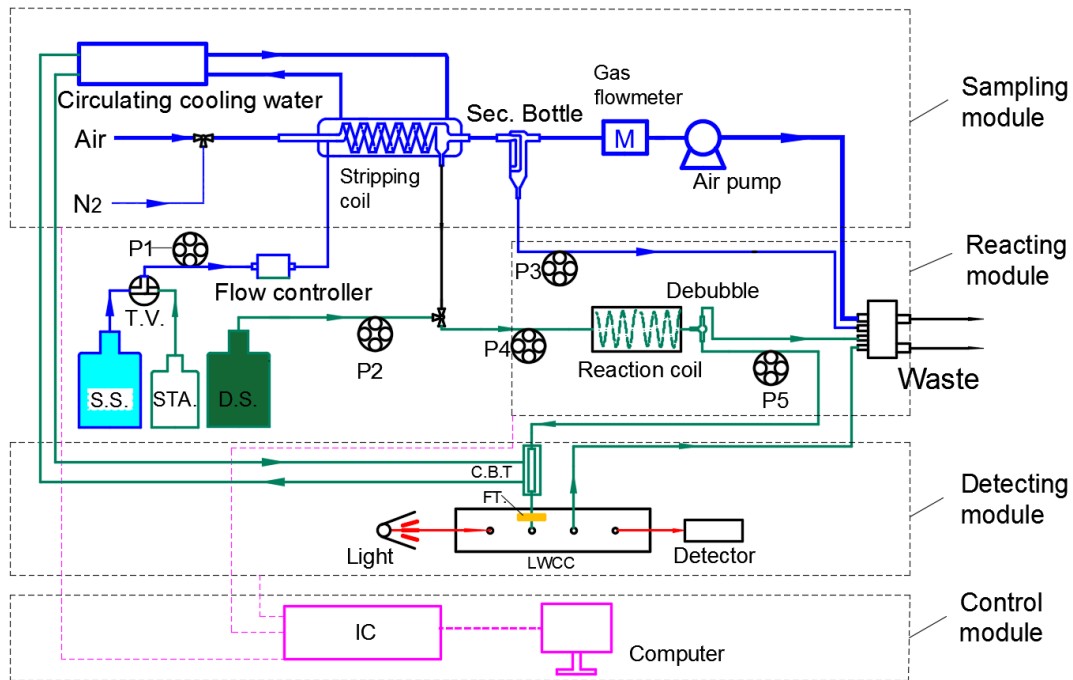


Fig. 1. Schematic diagram of SAC-LOPAP. (M: gas flowmeter; S.S.: stripping solution; STA.: standard solution;
D.S.: derivatization solution; Sec. Bottle: security bottle; C.B.T: cooling buffer tube; T.V.: triple valve for switching
stripping solution and standard solution; P1, P2, P3, P4, P5: peristaltic pump for transferring solutions; FT: syringe
filter; IC: integrated circuit. The gas flow rate can be controlled from 0.2-2.0 L min⁻¹, with an optimal gas flow rate
of 0.7 L min⁻¹. The liquid flow rate can be controlled from 0.1-1.0 ml min⁻¹, with an optimal stripping liquid flow
rate of 0.49 ml min⁻¹).
## 2.3 Experiment protocol
The exact recipe of the chemical reactions follows the reactions described by Searle et al (Searle and
Phillip, 1984). We used 0.75g L⁻¹ salicylic acid (TCI, 99.5%, Japan), 0.014 g L⁻¹ sodium nitroferricyanide
(TCI, 99%, Japan), and 0.2 g L$^{-1}$ NaOH as stripping solution (R1). Then the 0.188ml L$^{-1}$ Sodium
hypochlorite (Aladdin, active chlorine10%, China) and 1.5 g L$^{-1}$ NaOH as derivatization solution (R2).
We acknowledge that based on a selective colorimetric reaction to form a highly absorbing reaction
product must be carried out under catalytic and alkaline conditions. Sodium nitroferricyanide is
recognized as a high-efficiency catalytic to increase the sensitivity of the Equation 4 (Krom and Michael,
1980; Searle and Phillip, 1984).
Calibrating the setup uses NH$_4^+$ standard solution produced by the National Institute of Metrology,
China. The standards are prepared shortly before use by NH$_4^+$ standard solution with R1 in volumetric
bottle and to use it right after it was ready. The ideal use cycle of R1 and R2 was half a month,   after
replacement with new R1 and R2 solutions and other instrument fittings, the instrument should be
recalibrated to ensure data quality.

## 2.4  Sampling method

The inter-comparison experiment was conducted at the College of Environment Sciences and
Engineering, Peking University, located within the 4th ring road in northern Beijing, China (39.59° N,
116.18°E). A commercial instrument Picarro G2103 analyzer (Picarro, US) used for atmospheric NH$_3$
measurement based on the CRDS method was deployed concurrently with SAC-LOPAP in the
comparison, which could be used to validate other instruments (Twigg et al., 2022). The experiment took
place from 15 September 2021 to 15 October 2021, with the instruments installed in a field container.
Two instruments shared an inlet and were deployed 2.5 m above the ground. A Polytetrafluoroethylene
(PTFE) filter (46.2 mm diameter, 2 μm pore size, Whatman, USA) is used in the front of the sample
module to remove ambient aerosols, which is placed into a round filter holder made of perfluoro alkoxy
(PFA). We changed the filter every day with the aim of avoiding uncertainties. After the filtration of the
aerosols, the sample gas flow is delivered into a 3.8 m long 1/4-inch Teflon tube, and a temperature-
controlled metal heating wire (set at 35 °C ±0.1 °C) is wrapped around the sample tube and covered with
thermo-isolation materials. We ran our instrument with an additional drag flow of 1.75 L min$^{-1}$ with aim
to ensure the ambient residence time was about 7.8 msec for all instruments. Data acquisition times were
different for the above instruments during the inter-comparison. The base reporting periods for Picarro
and SAC-LOPAP were 1 s and 30 s. For the purposes of comparison, data from the two instruments
presented in this section were averaged to 30 s. In addition, high purity $N_2$ as zero gas was injected into
the sampling tube and carried out every 7 days at the start and end of the campaign as well. The standard
air source comes from China Sichuan Zhongce Biaowu Technology Co., LTD. The quality management
system of the company conforms to the recognized standard in the Chinese industry (GB/T9001-
2016/ISO 9001:2015). The composition was ammonia (5.08 ppm) and nitrogen with the uncertainty was
2%. In the test, pure $N_2$ was used as the dilution gas to obtain the required concentration of ammonia
standard gas. Calibrations were performed using combinations of concentrations at 1.32, 4.95, 9.59,
17.90 and 54.96 ppb from the cylinder. In addition, 4.95 ppb and 54.96 ppb standard gas were injected
into the sample tube every 7 days after zero point. The field container was controlled at 25 °C ±1 °C to
reduce the impact of temperature fluctuations on measurement results.

## 3 Characterization and optimization

## 3.1 Sampling efficiency

$NH_3$ Standard gas of 54.96 ppb was used as the sample to be collected through two identical serial
stripping coils, and the concentration of liquid samples collected by the two stripping coils was measured
to calculate the capture efficiency. The calculation formula is as below.
$$\gamma_1 = \frac{c_1}{c_1 + c_2} \times 100\% \qquad\qquad (6)$$
Where, $\gamma_1$ denotes the collection efficiency of the first stripping coil, $c_1$ and $c_2$ denote the concentration
of $NH_4^+$ trapped in the first stripping coil and the second stripping coil, respectively.
The collection efficiency of $NH_3$ from the R1 reached more than 99% under different $c_{NaOH}$, $F_l$, and
$F_g$. Figure 2a and Figure 2b show that the $F_l$ and the $F_g$ had almost no influence on collection efficiency.
Figure 2c shows that $c_{NaOH}$ of 1.25 mmol $L^{-1}$ achieved the greatest collection efficiency in the R1 (99.9%).
Therefore, the $c_{NaOH}$ of 1.25 mmol $L^{-1}$ was selected as the R1 of the $NH_3$. And we selected $F_l$ as 0.49 ml
$min^{-1}$ and $F_g$ as 0.7 L $min^{-1}$ in order to achieve the required detection range in this study.

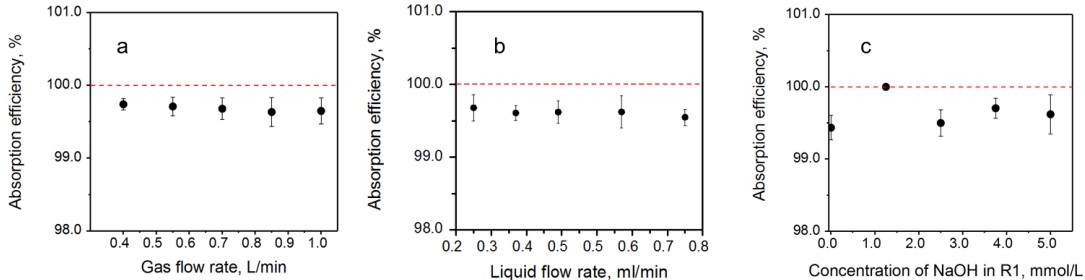


Fig. 2. The absorption efficiency of stripping coil versus (a) gas flow rate ($c_{NaOH}$ = 4.0 mmol $L^{-1}$, $F_l$ = 0.49 ml $min^{-1}$), (b) liquid flow rate ($c_{NaOH}$ = 4.0 mmol $L^{-1}$, $F_g$ = 0.7 L $min^{-1}$), (c) concentration of NaOH in R1 ($F_l$ = 0.49 ml $min^{-1}$, $F_g$ = 0.7 L $min^{-1}$).

## 3.2 Setting reaction conditions

However, precipitates can attach to the wall of the pipeline and LWCC for on-line instruments, which leads to pipeline blockage and baseline drift. Therefore, we need to optimize reaction conditions, add the constant temperature module and liquid flow controller temperatureto achieve continuous online measurement of low-concentration ammonia in ambient air. The concentration of the R1 we used in the initial reaction conditions (longer optical path and smaller sampling volume) contained 1 g $L^{-1}$ salicylic acid, 0.1 g $L^{-1}$ sodium nitroprusside, and 1 g $L^{-1}$ NaOH. 0.5 ml $L^{-1}$ sodium hypochlorite and 3 g $L^{-1}$ NaOH were used as R2 (Krom and Michael, 1980; Searle and Phillip, 1984). In addition, the syringe filter was introduced to minimize the influence of precipitate (Bianchi et al., 2012), but a large drift of the baseline would still occur during the long time run in our experiment, which will be discussed in detail later. In fact, we tried interrupting the sampling for a few minutes and implementing 5% hydrochloric acid for the system to remove these precipitates. However, the concentration changed greatly before and after each cleaning precipitation. In addition, once the precipitation was formed, it will take a long time to remove the precipitation, which will also increase the risk of contaminating the detector. According to reaction kinetics, reducing the stripping and derivatization concentrations (solution concentration) and [$OH^-$] of the system can greatly reduce the formation of precipitates in the solution. Therefore, we need to find the optimal reaction conditions to produce the least amount of precipitate. The maximum absorbance of a 100 μg $L^{-1}$ $NH_4^+$ standard solution was obtained at 18.75 mmol $L^{-1}$ $OH^-$ and we could obtain a high absorbance of light and a slow speed of precipitate formation, which meant that 1.5 g $L^{-1}$

NaOH was added to the derivatization solution, resulted in the precipitate in the solution being too small
to cause pipeline blockage and baseline drift. Importantly, we added regular assessment of the system
drift through use of online sampling of pure $N_2$. The range of blank signal in continuous operation for 48
h were 2.856 V ~ 2.848 V and 2.254 V ~ 1.834 V of reduced solution concentration and former high
solution concentration, and the maximum offset were 0.3% and 18.6%, respectively, the baseline of low
concentration solution has better stability (Fig. 3). In addition, the concentrations of salicylic acid, sodium
nitroferricyanide and sodium hypochlorite were 0.04, 0.02 and 0.006 times lower than those in previous
research, respectively (Bianchi et al., 2012). In general, the iron-containing precipitate increase the
absorbance by scattering or absorbing light, resulting in measurement bias. In this study, the amount of
iron-containing precipitation is very small by reducing the content of components and alkali of the
solution system, and the voltage of the instrument will not drop significantly due to contamination, which
is conducive to better maintenance of the baseline.

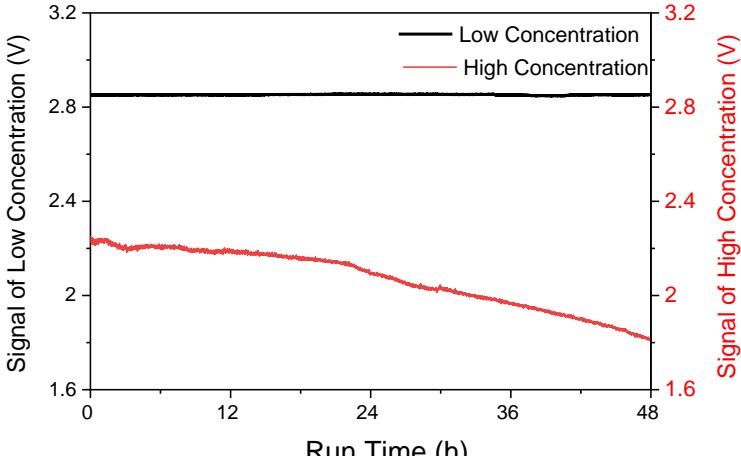


Fig. 3. The blank time series of the $NH_3$ detector ran continuously for 48 h.(Low concentration: 0.75g $L^{-1}$ salicylic
acid, 0.014 g $L^{-1}$ sodium nitroferricyanide, and 0.2 g $L^{-1}$ NaOH as R1, then the 0.188ml $L^{-1}$ Sodium hypochlorite and
1.5 g $L^{-1}$ NaOH as R2; High concentration: 1g $L^{-1}$ salicylic acid, 0.1 g $L^{-1}$ sodium nitroferricyanide, and 1 g $L^{-1}$
NaOH as R1, then the 0.5ml $L^{-1}$ Sodium hypochlorite and 3 g $L^{-1}$ NaOH as R2).

## 232 3.3 Stability of liquid flow and temperature

The temperature control module and flow control system were designed because of the sensitivity of
molecular absorption spectrophotometry to ambient temperature and residence time. A commercial PID
temperature controller was used to control the temperature of the reaction coil with the accuracy of ±0.1
℃. The temperature control module was used to control the constant temperature from the reaction coil
to LWCC at 55.0±0.1 ℃. At the same time, the flow control system could control the rotational speed of
the peristaltic. This system used a commercialized liquid flow meter (SLI-1000, Sensirion, Switzerland)
detect the flow rate and feedback to the peristaltic pump control by detecting the flow of tiny bubbles,
which further improved the stability of the reaction process. In other words, the flow control system
could avoid the flow rate dropping caused by the abrasion of the pump tube and increase the flow rate
caused by the replacement of the pump tube, keeping the R1 flow at a constant set point (0.49 ml min$^{-1}$).
In addition, we designed a buffer tube with a cooling function to further reduce the effects of
precipitation. After the derivatization reaction in the reaction coil at 55.0 ℃, the mixed solution entered
the cooling buffer tube. Most of the precipitation was generated in the buffer tube and attached to the
tube wall, while some of the precipitation generated in the downstream pipeline was intercepted by an
in-line precipitate filter with a pore size of 1.0 μm before the LWCC, and the filter was changed weekly.
Overall, the above work can make the instrument maintain a relatively stable reaction time and
temperature, which can promote a relatively stable reaction process, resulting in a high reproducibility
to the same concentration of NH$_3$. In the calibration process, R1 was used as diluent, and the
concentrations were 10, 25, 50, 75, 100, 150, and 200μg L$^{-1}$ of NH$_4^+$ standard solution. High purity N$_2$
was used as blank gas into the sampling tube, and the standard solution entered the solution system
instead of the R1. Fig. 4 showed the calibration with the NH$_4^+$ concentration gradient of 0, 10, 25, 50 and
100 μg L$^{-1}$ (150, and 200μg L$^{-1}$ of NH$_4^+$ standard solutions were out of the detection range, which was
discussed in section 3.4). Each concentration point was run for 40 minutes, and the relative standard
deviation (RSD) calculated from four consecutive measurements (the collection of the four replicates
were completed during a 4-week of constant instrument operation) ranged from 0.32 % to 2.65 %, with
the k varying from 0.0037 to 0.0040. Moreover, the blank experiment tests were automatically made
every one or two days, that is, high purity N$_2$ was used as a blank gas through the sample tube for 40
minutes, the RSD of the blank signal in continuous operation for one month was 1.8 %, which indicated
good repeatability and stability of the instrument. Seven switching samples were performed with 50 μg
L$^{-1}$ NH$_4^+$ standard solution and R1, after calculating 10-90 % of the full signal after a change in
concentration, the time response was approximately 140 s, which was much quicker than the method
described by Bianchi et.al (measured to be 10 min) (Bianchi et al., 2012).

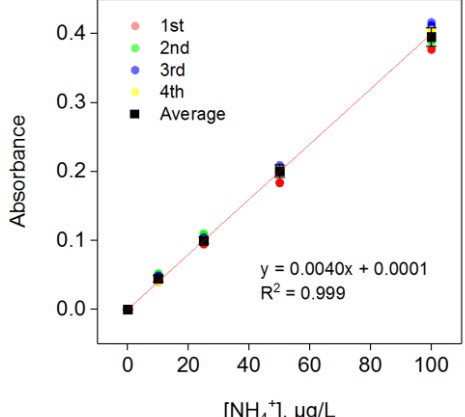


Fig. 4. Calibration curves of standard solution with the same concentration gradient 4 times
Table 1. Linear regression with the same concentration gradient 4 times

| Time | k | b | $R^2$ |
|------|------|------|------|
| 1st | 0.0037 | 0.0018 | 0.9998 |
| 2nd | 0.0039 | 0.0046 | 0.9996 |
| 3rd | 0.0040 | 0.0034 | 0.9997 |
| 4th | 0.0040 | 0.0003 | 0.9999 |

## 268 3.4 Setup of the temperature

High temperature can accelerate the reaction process and achieve better measurement accuracy and
precision. The voltage signal decreased with increasing temperature; conversely, the absorbance
increased with temperature. According to the flow rate (gas flow rate of 0.70 L min$^{-1}$, liquid flow rate of
0.49 ml min$^{-1}$), the detection limit of our SAC-LOPAP can reduce to less than 50 ppt when the absorbance
of 50 µg L$^{-1}$ NH$_4^+$ standard solution reached 0.15 or more. However, if the temperature is too high, there
is a danger that the pipeline interface of the instrument will fall off. Considering the continuous delivery
of solutions (the stability of pipeline connections) and the detection limit (lower than 50 ppt), 55 ℃ was
selected as the best reaction operating temperature of the instrument, at which sufficient absorbance could
be achieved to detect concentrations of ammonia gas. The standard solution entered the solution system
instead of the stripping solution, then the measured absorbance values were used as absorbance-standard
solution concentration plot and regression calculation (The experimental process has been described in
Section 3.3). The result is shown in Fig. 5, a high degree of correlation was found between the standard
solution and absorbance with a correlation coefficient of $R^2 = 0.99$ for the standard solution of 0-100 μg
$L^{-1}$, however, due to the incomplete reaction of $NH_4^+$ with dye products, there are two points outside of
the linear fit (standard solution concentrations are 150 and 200 μg $L^{-1}$). Therefore, the approximate
mixing ratio of $NH_3$ corresponding to the standard liquid concentration is 0-99.1 ppb, which is more
than adequate for monitoring urban areas. The detection limit for $NH_4^+$ liquid solution is about 40.9 ng
$L^{-1}$, which is calculated as 3 times the average standard deviation of blank signal noise in one hour. With
an air sample flow rate of 0.7 L $min^{-1}$ and a liquid flow rate of 0.49 ml $min^{-1}$, this translates to a gas phase
mixing ratio of about 40.5 ppt. In other words, the measurement range was 40.5 ppt up to 99.1 ppb for
$NH_3$, which was well suited for the investigation of the $NH_3$ budget from urban to rural conditions in
China. At the same time, according to the zero point data and the calibration, the corresponding
concentration to the voltage signal of 0.1 mV is 3.1 ppt, which far meets our requirements for actual
environmental measurement. Importantly, the detection limit can be decreased by improving the gas flow.
We can increase our detection range by reducing the reaction temperature and shortening the length of
LWCC. For example, the following table could be obtained according to Formula 5 and the stability
ranges of $F_l$ and the $F_g$. The detection limit could be reduced to 14.47 ppt and the detection upper limit
can be increased to 519.02 ppb by adjusting the $F_l$ and the $F_g$ (Table.2).

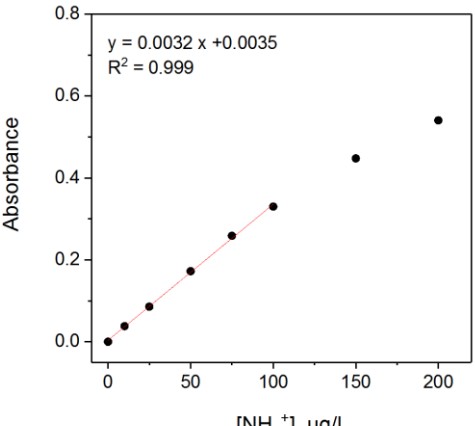


299                    Fig. 5. Standard solution and absorbance liner range test, to get a measurement range

300                    Table 2. Relationship between $F_l$ , $F_g$ and detection range of SAC-LOPAP

| $F_l$, ml min$^{-1}$ | $F_g$, L min$^{-1}$ | $C(NH_3)_{min}$, ppt, $[NH_4^+] = 40.9$ ng L$^{-1}$ | $C(NH_3)_{max}$, ppb, $[NH_4^+] = 100$ μg L$^{-1}$ |
|---|---|---|---|
| 0.25 | 1 | 14.47 | 35.38 |
| 0.35 | 0.85 | 23.84 | 58.29 |
| 0.5 | 0.7 | 41.35 | 101.11 |
| 0.75 | 0.4 | 108.55 | 265.41 |
| 1.1 | 0.3 | 212.28 | 519.02 |

## 4. Comparison in urban Beijing

The time series of the concentration of NH$_3$ during the inter-comparison period of Picarro and SAC-LOPAP were presented in Fig. 6a. There were a few data gaps for the above instruments caused by calibration operations and instrument maintenance. Instruments display similar temporal features for NH$_3$ concentrations over the duration of the study. In this study, the concentration of our instrument ranged from 1.3 ppb to 47.86 ppb with an average of 12.64 ± 8.63 ppb, which was close to the concentrations of Picarro (12.76 ± 8.57 ppb). The response speed was similar, indicated that SAC-LOPAP responded in time to rapid changed in NH$_3$ concentration. The diurnal variation results showed that the concentrations measured by the two instruments were very similar, with our instrument slightly lower than Picarro by 0.72 ppb (Fig. 6b). Furthermore, relatively good correlations for the NH$_3$ data observed by these instruments were achieved over a large dynamic range of concentration with a slope of 1.00 and an R$^2$ of 0.96 (Fig. 6c). We found that most of the time there were good correlations between the two instruments within one day except for the data of 23th and 30th September. The regression slope for all the days with higher and lower slopes are 1.46 and 0.72, respectively. We performed in-situ testing of both systems with a cylinder, we produced NH$_3$ concentrations of about 1.32, 4.95, 9.59, 17.90 and 54.96 ppb. Fig. 6d showed regression analyses of the NH$_3$ standard gas concentrations obtained with the two instruments. The NH$_3$ concentrations measured by picarro and our instrument were strongly correlated, with a slope of 1.01 and an R$^2$ of 0.99.

In general, our instrument run relatively stable with the standard deviation of zero gas during the one month of observations being within 26 ppt (Picarro: 23 ppt), which was far below our detection limit. Furthermore, the drift of SAC-LOPAP and Picarro at 4.95 ppb were 3.5% and 2.8%, while the drifts of 54.96 ppb were 1.5% and 0.7%, which meant that our instrument could keep steady for a long time and

it could be used for the continuous online measurement of low concentration of ambient air. More
detailed inter-comparison for these NH₃ instruments will be analyzed in a future publication.

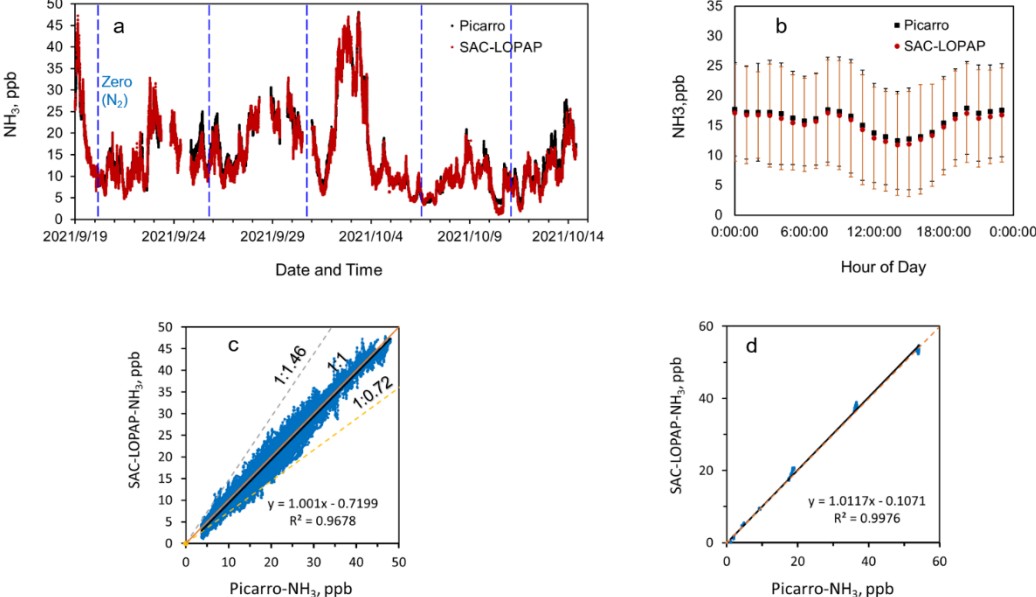

Fig. 6. (a) Time series of NH₃ concentration during the comparison, (b) Diurnal variation of NH₃ concentrations
observed by Picarro and SAC-LOPAP, (c) Regression analysis of the NH₃ concentrations observed by Picarro and
SAC-LOPAP, and (d) Regression analysis of different concentrations of Picarro and SAC-LOPAP NH₃ standard
gases.

## 5. Conclusions

Ammonia (NH₃) in the atmosphere affects the environment and human health and is therefore
increasingly recognized by policy makers as an important air pollutant that needs to be mitigated. The
accurate and precise detection of ambient NH₃ concentrations is therefore an urgent need for the
exploration of secondary pollution at the regional scale in China.

335       At the present stage, ambient NH₃ measurements at many supersites are still done with spectroscopic,

mass spectrometric and wet chemical methods, which are restricted by the high detection limit and lower
time resolution. In this study, we provide an online NH₃ monitoring system based on wet chemistry
stripping and long path absorption photometer of atmospheric NH₃, our new SAC-LOPAP system has
several significant improvements: one is the optimization of reaction conditions. The low concentration
but higher flow rate of solutions decreases the precipitate's production, and the cooling buffer tube and
the filter trap most of the precipitates. The others are the constant temperature module and liquid flow
controller. The constant temperature module in the system reduces the influence of ambient temperature
on the reaction process and color degree. Similarly, adding a liquid flow controller is helpful to the
stability of the flow rate and further increases the stability of the reaction process. These improvements
reduce the system error and significantly increase the sustainability of SAC-LOPAP operation. Our
instrument reached a detection limit of about 40.5 ppt with a stripping liquid flow rate of 0.49 ml $min^{-1}$
and a gas sample flow rate of 0.70 L $min^{-1}$ in the current condition, and the measuring range of the
instrument is 40.5 ppt to 99.1 ppb. Our system has also been characterized in a laboratory setting where
we can measure low concentrations. SAC-LOPAP and Picarro were compared in urban areas for a month
with relatively good agreement ($R^2$ = 0.967). In addition, the diurnal variation results showed that the
concentrations of the two instruments were very similar. Therefore, we conclude that our update of the
ammonia measurement experimental framework has been successful. However, more research about
field measurement and comparison is needed to verify the equipment's performance in routine
observation, and the influence of particulate ammonium on the results of $NH_3$ detection also requires
further study.

**Data availability.** The datasets used in this study are available from the corresponding author upon
request (hbdong@pku.edu.cn).

**Author contributions.** H.B.D. designed the study. S.S.T., K.X.Z. set up and characterized the instrument,
analyzed the data and wrote the paper with the input of H.B.D. As co-authors, S.S.T and K.X.Z.
contributed equally to this paper. All authors contributed to the field measurements, discussed and
improved the paper.

**Competing interests**. The authors declare that they have no conflict of interest.

**Acknowledgments**. This work was supported by special fund of State Key Joint Laboratory of
Environmental Simulation and Pollution Control (Grants No.22Y04ESPCP)

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
