# Peer review of "Colorimetric derivatization of ambient ammonia (NH3) for detection by"

_Atmospheric Measurement Techniques, 2023_

## Referee Comment (RC2)

**Review of "Improvement of online monitoring technology based on the Berthelot reaction and long path absorption photometer for the measurement of ambient NH₃: Field applications in low-concentration environments", by Tian (2023)**

The authors report the development of a new technique to measure atmospheric ammonia ($NH_3$) using wet chemistry based on the Berthelot reaction and a long path absorption photometer (SAC-LOPAP). The manuscript provides detail on the optimization of reaction conditions within the sampling and reacting modules, as well as a 1-month field evaluation with a co-located commercial Cavity Ring Down Spectroscopy instrument (CRDS). Given challenges inherent to measuring $NH_3$, the authors should be commended for developing a new $NH_3$ measurement technique. However, the manuscript needs major revisions to refocus the paper so that the introduction and conclusions are aligned with the results. Furthermore, additional information is needed in some places of the manuscript. General and specific comments are below.

General Comments:

The title and parts of the Introduction mention the low detection limit of the SAC-LOPAP and its ability to measure $NH_3$ in low-concentration environments. However, no detail is given on how the limit of detection (40.5 ppt) was quantified. Since the manuscript discusses how the low detection limit is one of the advantages of the SAC-LOPAP, there needs to be a detailed description of how the detection limit was calculated.

The field application of the SAC-LOPAP was not performed in a low $NH_3$ environment, with mixing ratios ranging from ~2 ppb to ~45 ppb. The title should be revised to clarify the field application was in an urban environment (i.e., Beijing) and not in a low $NH_3$ environment. Furthermore, the discussion throughout the manuscript should be reframed so that it focuses on the good agreement with the CRDS. Currently, portions of the manuscript focus on the SAC-LOPAP's ability to measure low concentrations and its low detection limit, neither of which have been shown or adequately explained.

Specific Comments:

Line 20 – state what the "established system" is in the abstract.

Line 21 – state what the "good correlation" is, as well as other relevant statistical metrics.

Line 38 – this should read "sulphuric acid" instead of "sulfate"; also, please define CLOUD.

Lines 42-43 – clarify that this study is specific to China, and please verify that the citation is correct.

Line 62 – define DFB.

Line 66 – what is meant by "$NH_3$ species"? This should be "$NH_3$", "$NH_x$ species", or "$NR_x$ species".

Lines 67-68 – what does "special materials" mean?

Lines 69 and 73 – there appear to be in-text citations errors, please correct/verify.

Lines 75-76 – discuss the importance of inlet design on ambient measurements of $NH_3$ and detection limits.

Line 89 – clarify what "statically" means and quantify what a "long time" is (e.g., months, years?).

Lines 92-93 – be quantitative with what is meant by "low-concentration" and "low detection limit".

Line 104 (Figure 1) – verify that reactants and products balance in the reaction scheme (e.g., the HCl appears to be missing from Step 2).

Line 111 – clarify what "$NH_3$ components" means.

Line 125 – correct "invert" to "convert" and provide the units for $C_{NH4+}$.

Line 126 – what is meant by gas "production sample"?

Lines 129-130 – should the temperature be for ambient air since the equation was derived by the Ideal Gas Law?

Lines 131-132 – please provide detail on the capture efficiency parameter. How was it determined, what does it depend on, and how sensitive is it to the set-up (e.g., temperature of the solution, pH of the solution, inlet design)?

Line 143 – rather than say "and so on", list of all the measures needed to achieve continuous online measurement.

Line 150 – should this be precipitate instead of sediment?

Line 156 – at a pH of ~12 and temperature of 55°C, gaseous ammonia won't be as soluble as under typical wet chemistry methods. What effect do the high pH and high temperature on the capture efficiency of ammonia?

Line 172 (Figure 4) – Define "high" and "low" concentration.

Lines 185-186 – Describe in detail how the detection limit was determined.

Lines 190-191 – Describe in detail how the upper range was determined.

Line 215 – at a pH ~12 most of the $NH_x$ would $NH_3$, and not $NH_4^+$.

Line 222 (Figure 6) – describe what approximate mixing ratio of $NH_3$ the calibration concentrations correspond to.

Line 233 – did the CRDS and SAC-LOPAP share an inlet, or did each have its own inlet at the same height?

Lines 249-251 – recommend including other statistical metrics in addition to $R^2$, such as mean bias or slope of linear regression.

Line 261 – most of the Conclusions section describes improvements made to the Berthelot reaction conditions, and do not necessarily represent an improved methodology for quantifying ambient $NH_3$ relative to other $NH_3$ measurement technologies. This sentence should be rephrased to clarify this nuance.

Lines 272-273 – as noted above, there was insufficient explanation for how the detection limit was determined.

---

## Author Comment (AC1)

Response to Referees

Manuscript Number: amt-2023-33

Manuscript Title: Improvement of online monitoring technology based on the Berthelot reaction and long path absorption photometer for the measurement of ambient NH₃: Field applications in low-concentration environments

The discussion below includes the complete text from the referees, along with our responses to the specific comments and the corresponding changes made to the revised manuscript.

The detailed answers to the individual referee's comments in blue.

All of the line numbers refer to the original manuscript.

Response to Referee #1 Comments:
* * *
We would like to thank the referee for his/her detailed comments and suggestions which helped us a lot to improve the quality of the paper. Our revised manuscript has been further edited by professional language services.

The Authors present a new colorimetric reaction to convert NH3 to a highly absorbing dye, taking advantage of the design principles of the LOPAP technique used extensively for HONO measurements. The technique is optimized for flows, reagent concentrations, other technical considerations, and then compared against an established optical technique in a field inter-comparison. Generally speaking, the basics of a good paper are present in this work. Technically speaking, the attention to detail and quality of the work are far below the expectations required for publication in Atmospheric Measurement Techniques. There are numerous flaws throughout the manuscript that need to be addressed by the addition of more data and/or the conduct of further experiments. The organization of the manuscript makes it almost impossible to follow what was done and why. The methods section is missing a large number of critical details around the instrument design and experiments reported throughout the results and discussion. The results and

discussion fail to place the technique in the context of the extensive literature on instrument intercomparisons for NH3 measurements, but also the general expectations from measuring NH3 in an urban atmosphere. The introduction fails entirely to convince one that the literature has been suitably surveyed to justify the creation of this new tool. Meanwhile, the resulting atmospheric observations are interesting and compare well to the established CRDS technique, although the interpretation and quality of the comparison are quite poor. Overall, this manuscript is currently completely unsuited for publication in AMT.

Major Comments

1. Introduction: Requires a complete re-write, based on the technical comments below. Most basic techniques and metrics for making atmospheric $NH_3$ measurements are surveyed through a single reference, if at all. Prior intercomparisons between techniques are not suitably referenced. Last, the chemistry for the conversion of NH3 to a highly absorbing dye is notably missing from the introduction. Given the long standing knowledge on this reaction, the benefits and drawbacks, as well as the typical samples that it has been applied to should be presented in detail.

2.Methods: The instrument description and supporting figure are in no way acceptable for publication. It would be impossible to reproduce this instrumentation with the level of detail provided. Even the general components of the instrument are not communicated in the written section, or in the figure caption.

This section should also contain all the information about the chemicals or solvents used, how the various solutions were made and at what concentration, details on the purity/suppliers, and so on.

The experimental details of the various sections of the results and discussion are also missing from the methodology and should have some description here, along with the basic motivation for each variable that had to be optimized. Flows in particular are very poorly outlined, which is surprising since so much effort was put into a recursive algorithm to autocorrect for degradation of peristaltic pump tubing.

Last, the details of the field measurements need to be relocated here, alongside the experiments conducted in the field for conducting blanks and positive controls (NH3 additions).

3. Results and Discussion: Many of the tables and figures can be replaced with a single sentence to communicate the findings. These are low quality and should be removed. There are a number of figures for which the Authors likely have data, to demonstrate the performance of their instrument, which are not presented. Some tables and figure panels are presented and never discussed in the text. Substantial attention and revision throughout the results and discussion are required.

4.Results and Discussion: The intercomparison is the capstone of this work and the quality of the analysis here requires a complete overhaul. The literature (and textbooks) are replete with examples of how to obtain a lot of insight into issues with one or both instruments during an intercomparison. Not only have the Authors neglected the basic presentation of the regression, slope, and intercept, but there is also no discussion around what these findings imply based off of the hundreds (if not more) of prior reports doing the exact same kind of comparison between two analytical methodologies. This is a fatal flaw and must be corrected. Further to this, attention to the fundamental basics of reporting analytical data is not made. For example, the Authors report detection limits (without any description of how they did the calculation) of 40.5 ppt in Section 3.2 and then proceed to report measured blank values below this in Table 2. The reported detection limit has more significant digits than any of the reported blank values as well, which suggests that this concept throughout is not being properly reviewed.

We thank the reviewer for taking the time to review this study and providing the constructive feedback of the manuscript.

Technical Comments (line by line):

Lines 1-3: The title is confusing and long. It uses a named reaction, which is not really relevant to what is being presented in the manuscript. The reaction is one that is very long established, but has no bearing on the analytical approach. Basically, a highly UV-Vis absorbing dye is obtained from an analyte which itself is a poor absorber. This is then coupled with a very long pathlength optical sample cell, to minimize the amount of analyte that can be detected, through molecular absorption methods via the Beer-Lambert Law (which is conspicuously not presented). Named reactions are outdated ways of organizing methods, which are not accessible to an atmospheric science

community composed of chemists, engineers, earth scientists, and physicists (amongst many others). Reference to a named organic chemistry reaction simply reduces the impact and uptake of this work. Consider revising the title to something more concise: Colorimetric derivatization of ambient ammonia ($NH_3$) for detection by long path absorption photometry.

Many thanks to the reviewer for the proposed title. The "Improvement of online monitoring technology based on the Berthelot reaction and long path absorption photometer for the measurement of ambient $NH_3$: Field applications in low-concentration environments" has been revised as "Colorimetric derivatization of ambient ammonia ($NH_3$) for detection by long path absorption photometry" according to the journal guidelines.

Line 13: SAC is not defined at first use, which is required according to the journal guidelines.

The "SAC" has been revised as "Salicylic acid chromogenic and long path absorption photometer (SAC-LOPAP)" according to the journal guidelines.

Lines 13: 'Berthelot reactions' - Named reactions are often highly colonial and exclusionary. Suggest replacing all references to this reaction with 'based on a selective colorimetric reaction to form a highly absorbing reaction product' or something equivalent throughout the remainder of the manuscript. It would also be useful if the authors comment on the strength of UV-Vis absorption that NH4+ has compared to the indophenol blue. The only place the named reaction really is relevant is in the caption for the reaction scheme, where it can be pointed out that this reaction has been known for ~150 years.

The "Berthelot reactions" has been revised "based on a selective colorimetric reaction to form a highly absorbing reaction product" as suggested.

Line 14: 'run statically' – What do the Authors mean by this? That the instrument is stationary while operating? Is this something that needs to be stated? Or do they mean something else? Revise for clarity.

The "run statically" refers to the instrument does not fluctuate greatly due to the precipitate. The "run statically" has been revised "run stably".

Line 17: 'under stable conditions' – Do the Authors mean 'under optimal conditions'? A final optimal method is never clearly stated anywhere in the results and discussion. Revise for clarity here and below.

Yes, the "run statically" refers to under optimal conditions. we revised the "under stable conditions" as "with a stripping liquid flow rate of 0.49 ml min$^{-1}$ and a gas sample flow rate of 0.70 L min$^{-1}$", and we stated the final optimal method in section 3.1.

**3.1 Sampling efficiency**

NH$_3$ Standard gas of 54.96 ppb was used as the sample to be collected through two identical serial stripping coils, and the concentration of liquid samples collected by the two stripping coils was measured to calculate the capture efficiency. The calculation formula is as below.

$$\gamma_1 = \frac{c_1}{c_1+c_2} \times 100\% \tag{6}$$

Where, $\gamma_1$ denotes the collection efficiency of the first stripping coil, $c_1$ and $c_2$ denote the concentration of NH$_4^+$ trapped in the first stripping coil and the second stripping coil, respectively.

The collection efficiency of NH$_3$ from the R1 reached more than 99% under different $c_{NaOH}$, $F_l$, and $F_g$. Figure 2a and Figure 2b show that the $F_l$ and the $F_g$ had almost no influence on collection efficiency. Figure 2c shows that $c_{NaOH}$ of 1.25 mmol L$^{-1}$ achieved the greatest collection efficiency in the R1 (99.9%). Therefore, the $c_{NaOH}$ of 1.25 mmol L$^{-1}$ was selected as the R1 of the NH$_3$. And we selected $F_l$ as 0.49 ml /min and $F_g$ as 0.7 L min$^{-1}$ in order to achieve the required detection range in this study.

[Figure]

Fig. 2. The absorption efficiency of stripping coil versus (a) gas flow rate ($c_{NaOH}$ = 4.0 mmol L$^{-1}$, $F_l$

= 0.49 ml min$^{-1}$), (b) liquid flow rate ($c_{NaOH}$ = 4.0 mmol L$^{-1}$, $F_g$ = 0.7 L min$^{-1}$), (c) concentration of NaOH in R1 ($F_l$ = 0.49 ml min$^{-1}$, $F_g$ = 0.7L min$^{-1}$).

Line 18: 'varied from background contamination' – What do you mean? The background (or blank) experiments were contaminated and set the lowest value you could measure? Based on what seems to be presented in the results and discussion, the Authors have evaluated simple reagent blanks to determine instrumental detection limits instead of method detection limits (i.e. those that include sampling clean air). This may be incorrect word selection, but overall this does not seem relevant here and background contamination should be corrected for through the conduct of the regular measurement of negative controls (which are another outstanding issue in the results and discussion) delivered to the sampling inlet.

We agree with the referee that this is incorrect word selection. We deleted this sentence.

Line 18: 'in the current condition' - What does this mean? In your ambient measurements? Or using the optimized parameters that follow? If you mean the latter, then get those details out of brackets and make this sentence clearer.

Yes, the "in the current condition" refers to using the following optimization parameters. We get those details out of brackets as suggested.

Line 20: 'with another established system' – Be specific and state that you are comparing to a commercial cavity ring down spectrometer here.

The "with another established system" has been revised "a commercial instrument Picarro G2103 analyzer (Picarro, US)" as suggested.

Line 21: 'the two instruments had good correlation' - Qualitative. This is the focus of the paper. Slope, intercept, and R2 are all necessary for the abstract.

The "the two instruments had good correlation" has been revised "two instruments had good correlation with a slope of 1.00 and an R$^2$ of 0.96" as suggested.

Line 35: It does not react with the precursors, it reacts with the acid products. Revise for correctness.

We agree with the referee. We revised this sentence as "$NH_3$ easily forms ammonium ions ($NH_4^+$) with water and reacts with acidic species to form secondary inorganic particles".

Lines 37 and 39: Referencing is completely incorrect. The second instance is a very bad example of writing about another scientist's work and citing poorly/repetitively. Revise for all citation issues throughout to meet journal guidelines.

We revised all reference formatting as suggested.

Line 44: How is there no discussion on the sources and sinks of atmospheric $NH_3$ in the gas phase, nor its environmental impacts? The Authors also fail to report the typical range of mixing ratios of NH3 found in various atmospheric environments, or basic controls on its abundance (e.g. diurnal or seasonal trends). The prior paragraph only discusses particles and even that is poorly reviewed from the literature. Revise.

We agree with the referee and we revised this text "These secondary particles are considered a major source of fine particulate matter (PM), which is harmful to climate, visibility and human health (Wang et al., 2015). Furthermore, recent studies have shown that $NH_3$ is necessary to control fine particulate pollution (Wen et al., 2018; Wang et al., 2013). Due to those problems, the inventory of $NH_3$ emissions and concentration in urban air has been highly evaluated. Agriculture, including animal feedlot operations, is considered as the largest emission source of $NH_3$ with 80.6% of the global anthropogenic emissions followed by 11% from biomass burning and 8.3% from the energy sector, including industries and traffic (Behera et al., 2013). Expert estimate that global annual emissions of $NH_3$ will increase from 65 Tg N $yr^{-1}$ in 2008 to 135 Tg N $yr^{-1}$ in 2100 (Fowler et al., 2015). However, ambient measurement of $NH_3$ concentrations is difficult due to several factors: ambient levels vary widely with from 5 pptv to 500 ppbv (Janson et al., 2010; Krupa, 2003; Sutton et al., 1995). Ammonia exists in gaseous, particulate and liquid phases, which add further complicates the measurement (Warneck, 1988). In addition, $NH_3$ is "sticky" and interacts with surfaces of materials, resulting in slow inlet response times (Yokelson et al., 2003). Finally, the temperature difference between the indoor and outdoor environments and the humidity difference

between the inside and outside of the instrument will reduce the accuracy of measurement and calibration".

Line 48: 'low' – Should be 'biased. The NH3 eventually comes off the surface. It is participating in a reversible dynamic equilibrium. Revise.

We agree with the referee and we deleted this sentence.

Line 51: Improper reference formatting in another instance that needs to be fixed throughout.

We revised all reference formatting as suggested.

Line 53: This motivation section is poor and needs to be re-written. There is extensive literature on these kinds of inlet effects and much of it can be found in AMT or ACP. Please conduct a better literature review and better represent the current state of scientific knowledge.

We agree with the referee and we re-wrote the motivation as follows: However, ambient measurement of $NH_3$ concentrations is difficult due to several factors: ambient levels vary widely with from 5 pptv to 500 ppbv (Janson et al., 2010; Krupa, 2003; Sutton et al., 1995). Ammonia exists in gaseous, particulate and liquid phases, which add further complicates the measurement (Warneck, 1988). In addition, $NH_3$ is "sticky" and interacts with surfaces of materials, resulting in slow inlet response times (Yokelson et al., 2003). Finally, the temperature difference between the indoor and outdoor environments and the humidity difference between the inside and outside of the instrument will reduce the accuracy of measurement and calibration.

Line 59: 'viscosity' - Incorrect. And this sentence is repeating the metal issue from above with basically an identical sentence. This is very poor writing. Please revise for clarity.

The sentence has been revised as "the "sticky" of $NH_3$ will affect background, detection efficiency and detection response time of the instrument".

Line 63: 'high detection accuracy and a low detection limit' - Vague and qualitative. Lots of other reports on the QCL out there that are not cited here. This work is not the seminal one.

We agree with the referee and we have referred to more scholars' literatures: Utilizing a quantum cascade laser (QCL) or a DFB laser in a near-infrared band as the light source can achieve a low detection limit of 0.018ppb (Whitehead et al., 2008; Mcmanus et al., 2002; Von et al., 2009)

Line 64: 'of low concentration' - If you do not report what they actually achieve (or what the atmospheric range actually is - another thing that has been missed in this introduction) then this is just an arbitrary statement. Revise.

We gave the detection limit "a low detection limit of 0.018ppb" as suggested, so that low concentrations in the environment can be measured

Line 66: What is an 'NH$_3$ species'? This terminology is incorrect. Revise.

The "NH$_3$ species" has been revised "NH$_3$".

Line 67: Again, there are many other much better CIMS reports out there. Do a more comprehensive literature review. This is a journal focused on atmospheric techniques and instrumentation. The review of those should be excellent. This is poor.

We agree with the referee and we have referred to more scholars' literatures: The chemical ionization mass spectrometer (CIMS) technique is based on an ion-molecule reaction to selectively ionize and detect traceNH$_3$ in the atmosphere, which features a fast response and in situ measurement (Benson et al., 2010; Nowak et al., 2007; Yu and Lee, 2012).

Line 67: 'The sensor measurement' - What sensor? Is this different from CIMS? Too vague. Revise.

We think the sentence is inappropriate and we deleted it.

Lines 70-71: Coated wet annular denuder. Again, the literature review is isolated to a single paper and has not drawn from the seminal works that established these techniques. There are many MANY out there that are coupled to either flow injection analysis (FIA) or ion chromatography (IC).

We agree with the referee and we have referred to more scholars' literatures: Wet chemistry methods convert gas-phase NH$_3$ to aqueous NH$_3$ (NH$_4^+$) for online analysis by means of online ion

chromatography with a detection limit of 0.05 μg m$^{-3}$ (0.72ppb at 25 ºC) (Khlystov et al., 1995; Dong et al., 2012; Makkonen et al., 2012).

Line 79: This is not 'furthermore'. Present the actual reaction from figure 1 (as a series of numbered reactions, as per many other reports in this journal), and its historic development here. It has been used for decades for routine derivatization and colorimetric analysis of NH$_4^+$ in a wide variety of environmental samples (e.g. soils, environmental waters, etc). These should also be clearly presented here.

We agree with the referee and we added this sentence "it has been used for decades for routine derivatization and colorimetric analysis of NH$_4^+$ in a wide variety of environmental samples (e.g. soils, environmental waters, etc)".

Line 86: If you are going to build a LOPAP for NH$_3$, perhaps you ought to describe what a LOPAP is and how it works in the introduction? Seems like a large oversight.

We briefly describe the characteristics of LOPAP here, which are described in more detail in Section 2.2, the corresponding sentence is revised as follows: In those studies the product was detected by a long- path absorption photometer (LOPAP), in which the absorbance of the solution is amplified in the long-l path module to reach a lower detection limit (Heland et al., 2001).

Section 2.2: After the chromogenic reaction, the sample is sent to the detecting module, which comprises a liquid waveguide capillary cell (LWCC-100, World Precision Instruments, USA) with optical path length of 100 cm, an LED light source with the mode at 665 nm (Ocean Optics)and a phototube (S16008-33, HAMAMTSU, Japan) for the long path photometry detection The sample solution to be tested is filtered by a 1.0 μm filter before passing through LWCC to avoid interference from components of the sample matrix/method reagents. Lines 83-90: This section of a manuscript usually provides a clear structure for the sections/experiments conducted throughout the rest of the work. Laying out the motivations, and logic connecting each, in order here will help the Authors re-organize the entire manuscript for clarity.

We re-arranged this part of the manuscript according to the suggest, the corresponding text is revised as follows: In this study, we provide an online $NH_3$ monitoring system based on wet chemistry stripping of atmospheric $NH_3$, followed by the formation of a highly light-absorbing indophenol after a salicylic acid a derivatization reaction to produce the colored reaction product reaction and detected with LOPAP. According to Lambert-Beer's law, the sensitivity of spectrophotometry can be enhanced by increasing the optical path length. This sensitive analytical method has already been successfully applied in different colorimetric detection studies (Yao et al., 1998; Heland et al., 2001; Callahan et al., 2002). In analogy to the original long path absorption photometer (LOPAP) which was developed for HONO measurements (Kleffmann et al., 2002), we call this monitoring system the salicylic acid a derivatization reaction to produce the colored reaction product and long path absorption photometer (SAC-LOPAP), which features several improvements over versions previously reported by other groups: one is the optimization of reaction conditions, the other modification is the use of constant temperature module and flow control system. Secondly, we will present measurements demonstrating our new system in urban environments in Peking University, with measure low concentrations, good stability and low detection limit.

Line 89: Same issue here as in the abstract. What do the Authors mean by 'statically' here? That it can be set up and left to run for long periods of time? Revise.

Yes, "statically" refers to that it can be set up and left to run for long periods of time stably for one months. We modified the text accordingly.

Line 98: Do not used names for reactions. Simply present the actual reaction and you can note that it is named in the caption for the reaction scheme, if you feel it is really important. Remember, your readers are not necessarily chemists. They could be engineers, atmospheric scientists, or established chemists who have not thought about named organic reactions in decades.

The sentence has been removed as suggested.

Line 104: The reactions in this 'figure' belong in the introduction to bolster the sections already noted. Should not be a figure. This is a reaction, so denote it as one. Each step can be presented as a separate reaction in sequence. There are plenty of examples of this throughout AMT and ACP.

The text has been updated as suggested.

Lines 106-107: Revise. Explain what the glass coil does then give it the name of stripping coil.

We rewrote the sentence as suggested: The key component of the sampling module is a glass coil reactor, which is an open glass tube (inner diameter 1.5 mm, 75 cm long) coiled 12 turns. At the beginning of this coil, there is a flow manifold to mix the ambient air flow and the stripping solutions. At the end of this coil, it is further connected to a gas–liquid separator.

Why are you not operating a second channel here to correct for background issues in the reagents and/or interferences from the gas samples, like the HONO or NO2 LOPAPs do? This approach is not even discussed in the results and discussion, to contextualize the limitations of this new instrumentation. This is another glaring oversight.

HONO mainly removes the interference of $NO_2$ in the air, while gaseous ammonia is very stable and there is no interference from other components in the atmosphere. The zero drift is mainly caused by precipitation, and the interference cannot be deducted with operating a second channel. In addition, once the precipitation was formed, it will take a long time to remove the precipitation, which will also increase the risk of contaminating the detector.

Line 112: 'higher absorption efficiency' - The experiments determining this from standard gas flows are never presented, nor the value used. A section where those are determined (this is one of many missing and high-utility figures) and reported must be added to the manuscript.

This data has been added to Section 3.1

**3.1 Sampling efficiency**

$NH_3$ Standard gas of 54.96 ppb was used as the sample to be collected through two identical serial stripping coils, and the concentration of liquid samples collected by the two stripping coils was measured to calculate the capture efficiency. The calculation formula is as below.

$$\gamma_1 = \frac{c_1}{c_1+c_2} \times 100\% \tag{6}$$

Where, $\gamma_1$ denotes the collection efficiency of the first stripping coil, $c_1$ and $c_2$ denote the

concentration of $NH_4^+$ trapped in the first stripping coil and the second stripping coil, respectively.

The collection efficiency of $NH_3$ from the R1 reached more than 99% under different $c_{NaOH}$, $F_l$, and $F_g$. Figure 2a and Figure 2b show that the $F_l$ and the $F_g$ had almost no influence on collection efficiency. Figure 2c shows that $c_{NaOH}$ of 1.25 mmol L$^{-1}$ achieved the greatest collection efficiency in the R1 (99.9%). Therefore, the $c_{NaOH}$ of 1.25 mmol L$^{-1}$ was selected as the R1 of the $NH_3$. And we selected $F_l$ as 0.49 ml /min and $F_g$ as 0.7 L min$^{-1}$ in order to achieve the required detection range in this study.

[Figure]

Fig. 2. The absorption efficiency of stripping coil versus (a) gas flow rate ($c_{NaOH}$ = 4.0 mmol L$^{-1}$, $F_l$ = 0.49 ml min$^{-1}$), (b) liquid flow rate ($c_{NaOH}$ = 4.0 mmol L$^{-1}$, $F_g$ = 0.7 L min$^{-1}$), (c) concentration of NaOH in R1 ($F_l$ = 0.49 ml min$^{-1}$, $F_g$ = 0.7L min$^{-1}$).

Line 114: 'an alkaline derivatization solution' - Vague. Be specific with all reagents throughout. Concentrations, composition, stoichiometry, etc. There should be an entire sub-section in the methods detailing all of this.

Section 2.3 has been added to the method detailing all of this as suggested.

2.3   Experiment protocol

The exact recipe of the chemical reactions follows the reactions described by Searle et al (Searle and Phillip, 1984). We used 0.75g L$^{-1}$ salicylic acid (TCI, 99.5%, Japan), 0.014 g L$^{-1}$ sodium nitroferricyanide (TCI, 99%, Japan), and 0.2 g L$^{-1}$ NaOH as stripping solution (R1). Then the 0.188ml L$^{-1}$ Sodium hypochlorite (Aladdin, active chlorine10%, China) and 1.5 g L$^{-1}$ NaOH as A derivatization reaction to produce the colored reaction product solution (R2). We acknowledge that based on a selective colorimetric reaction to form a highly absorbing reaction product must be carried out under catalytic and alkaline conditions. Sodium nitroferricyanide is recognized as a highefficiency catalytic to increase the sensitivity of the Equation 4 (Krom and Michael, 1980; Searle and Phillip, 1984).

Line 115: 'kettle' - Kettle is the wrong word to use here. These are typically coils of tubing wrapped around a solid material of some sort, so the reaction can proceed to completion. Sometimes also a heated compartment. A kettle is a large volume pot, which is not consistent with this or the writing that follows stating exactly this. Suggest revising to 'coil' throughout the manuscript.

The "kettle" has been revised as "coil".

Lines 119-120: 'an LED light source, and a photoelectric detector' – Must provide all relevant details of these so this can be reproduced. Otherwise, why are you even reporting that this instrument can be constructed if the community cannot go on to make more of them?

The "an LED light source, and a photoelectric detector" has been revised as "an LED light source with the mode at 665 nm (Ocean Optics) and a phototube (S16008-33, HAMAMTSU, Japan)".

Line 120: 'is filtered…' - I do not see this filter in the Figure 2 diagram. How often does this need to be replaced? More details come below, but they belong here. The figure caption needs to do a better job of identifying this easily for readers.

The image and caption have been corrected to help readers better identify the instrument as suggested.

[Figure]

Fig. 1. Schematic diagram of SAC-LOPAP. (M: gas flowmeter; S.S.: stripping solution; STA.: standard solution; D.S.: derivatization solution; Sec. Bottle: security bottle; C.B.T: cooling buffer tube; T.V.: triple valve for switching stripping solution and standard solution; P1, P2, P3, P4, P5: peristaltic pump for transferring solutions; FT: syringe filter; IC: integrated circuit. The gas flow rate can be controlled from 0.2-2.0 L min$^{-1}$, with an optimal gas flow rate of 0.7L min$^{-1}$. The liquid flow rate can be controlled from 0.1-1.0 ml min$^{-1}$, with an optimal stripping liquid flow rate of 0.49ml min$^{-1}$).

Line 121: 'contamination' - The precipitates contain your derivatization product? If yes, then this would be contamination. If no, then this is an interference from components of the sample matrix/method reagents. Use analytical chemistry terminology appropriately.

 The precipitates do not contain derivative product, and the "contamination" has been revised as "an interference from components of the sample matrix/method reagents".

Lines 121-122: If you made the circuit and touch panel, the details are insufficient for this manuscript. Requires its own detailed section that follows the description of the LWCC.

We didn't make the circuits and the touch screens ourselves, and we have removed this sentence and replaced it with "Both the fluid propulsion module and detection module can be computer controlled".

Line 122: 'LWCC is 100 cm' – Optical filter for wavelength selection? Any other critical optical components missing in this description? Based on benchtop versions of this for soil extracts, the overall description here seems to be highly limited and should be expanded so all aspects of the new instrumental innovation can be assessed.

LWCC is introduced as follows in this paper: After the chromogenic reaction, the sample is sent to the detecting module, which comprises a liquid waveguide capillary cell (LWCC, World Precision Instruments, USA) with optical path length of 100 cm, an LED light source with the mode at 665 nm (Ocean Optics) and a phototube (S16008-33, HAMAMTSU, Japan) for the long path photometry detection.

Lines 123-124: What is this standard solution used for? Calibration of the LWCC? If yes, create a separate subsection. This section is getting very disorganized at this point.

We have created a separate section to explain the standard solution as suggested.

2.3   Experiment protocol

Calibrating the setup uses $NH_4^+$ standard solution produced by the National Institute of Metrology, China. The standards are prepared shortly before use by $NH_4^+$ standard solution with R1 in volumetric bottle and to use it right after it was ready. After replacement with new R1 and R2 solutions and other instrument fittings, the instrument should be recalibrated to ensure data quality.

Line 128: 'content of the air sample' - Give units. There are a lot of ways to report gas phase concentrations out there. Seeing as most of the figures below report mixing ratios with units of ppb, then perhaps that is what should be given here (with appropriate modifications to E1)

The "Where $C_{NH_3}$ denotes the content of $NH_3$ in the air sample" has been revised as "Where $C_{NH_3}$ denotes the content of $NH_3$ in the air sample (ppb)".

Line 129: 'NH3 (g/mol)' – If you are going to give this information, then you might as well give the actual value used too.

We gave the actual value as suggested, the "g/mol" has been revised as "(17 g/mol)".

Line 132: '(a constant determined by laboratory)' – This is never presented, either the method, nor the results and determined constant. The Authors should be reporting the best value acheived here, along with a reference to the section that describes those experiments and the results/discussion section that contextualizes them.

 Chapter 3.1 was added to describe the parts and results of these experiments as suggested.

**3.1 Sampling efficiency**

NH$_3$ Standard gas of 54.96 ppb was used as the sample to be collected through two identical serial stripping coils, and the concentration of liquid samples collected by the two stripping coils was measured to calculate the capture efficiency. The calculation formula is as below.

$$\gamma_1 = \frac{c_1}{c_1+c_2} \times 100\% \tag{6}$$

Where, $\gamma_1$ denotes the collection efficiency of the first stripping coil, $c_1$ and $c_2$ denote the concentration of NH$_4^+$ trapped in the first stripping coil and the second stripping coil, respectively.

The collection efficiency of NH$_3$ from the R1 reached more than 99% under different $c_{NaOH}$, $F_l$, and $F_g$. Figure 2a and Figure 2b show that the $F_l$ and the $F_g$ had almost no influence on collection efficiency. Figure 2c shows that $c_{NaOH}$ of 1.25 mmol L$^{-1}$ achieved the greatest collection efficiency in the R1 (99.9%). Therefore, the $c_{NaOH}$ of 1.25 mmol L$^{-1}$ was selected as the R1 of the NH$_3$. And we selected $F_l$ as 0.49 ml /min and $F_g$ as 0.7 L min$^{-1}$ in order to achieve the required detection range in this study.

[Figure]

Fig. 2. The absorption efficiency of stripping coil versus (a) gas flow rate ($c_{NaOH}$ = 4.0 mmol L$^{-1}$, $F_l$

= 0.49 ml min$^{-1}$), (b) liquid flow rate ($c_{NaOH}$ = 4.0 mmol L$^{-1}$, $F_g$ = 0.7 L min$^{-1}$), (c) concentration of NaOH in R1 ($F_l$ = 0.49 ml min$^{-1}$, $F_g$ = 0.7L min$^{-1}$).

Line 134: The caption for Figure 2 is too simplistic. This caption should be defining all the shorthand notation above at a minimum. It could also contain information on the variables that can be controlled (e.g. flow rates of gas and liquids), as well as to depict their optimized values determined in this work. There are a lot of components that are not described anywhere (e.g. things here that I assume are peristaltic pumps, but they aren't labeled or described; or whatever the empty pink box is connected to the computer?) that should be in some kind of legend.

The caption for Figure 1 has been corrected as suggested.

[Figure]

Fig. 1. Schematic diagram of SAC-LOPAP. (M: gas flowmeter; S.S.: stripping solution; STA.: standard solution; D.S.: derivatization solution; Sec. Bottle: security bottle; C.B.T: cooling buffer tube; T.V.: triple valve for switching stripping solution and standard solution; P1, P2, P3, P4, P5: peristaltic pump for transferring solutions; FT: syringe filter; IC: integrated circuit. The gas flow rate can be controlled from 0.2-2.0 L min$^{-1}$, with an optimal gas flow rate of 0.7L min$^{-1}$. The liquid flow rate can be controlled from 0.1-1.0 ml min$^{-1}$, with an optimal stripping liquid flow rate of

0.49ml min$^{-1}$).

Lines 136-140: This does not belong here. It should be part of the methods section. None of this is shown in the reaction scheme, so its relevance is VERY unclear. How is this so important that it is the first thing that needs to be mentioned? Where is the Fe coming from? One of your reagents? I do not see any description of chemicals used, suppliers, purity, concentration, solvents, etc. provided in the methods section. These all need to be added.

This part has been put into part of the methods section as suggested.

2.3  Experiment protocol

The exact recipe of the chemical reactions follows the reactions described by Searle et al (Searle and Phillip, 1984). We used 0.75g L$^{-1}$ salicylic acid (TCI, 99.5%, Japan), 0.014 g L$^{-1}$ sodium nitroferricyanide (TCI, 99%, Japan), and 0.2 g L$^{-1}$ NaOH as stripping solution (R1). Then the 0.188ml L$^{-1}$ Sodium hypochlorite (Aladdin, active chlorine10%, China) and 1.5 g L$^{-1}$ NaOH as A derivatization reaction to produce the colored reaction product solution (R2). We acknowledge that based on a selective colorimetric reaction to form a highly absorbing reaction product must be carried out under catalytic and alkaline conditions. Sodium nitroferricyanide is recognized as a high-efficiency catalytic to increase the sensitivity of the Equation 4 (Krom and Michael, 1980; Searle and Phillip, 1984).

Do these precipitates have little effect because they can be washed out of the system? If yes, can't you simply automate an occasional washing step in this instrument?

In fact, this precipitate is not easily washed out of the system.   In addition, we tried interrupting the sampling for a few minutes and have implemented 5% hydrochloric acid for the system to remove these precipitates. However, the background voltage changes greatly with 100 mv before and after each cleaning, resulting in low measurement accuracy.

Lines 147-148: 'based on the former scholar' – These reagents are used to derivatize NH4+ in numerous standard procedures in monitoring labs. If the Authors want to cite some seminal works, that is fine, but they do not need to write that this knowledge originates from scholars.

The 'based on the former scholar' has been removed as suggested.

Lines 148-149: 'and the state… initial reaction condition' - Why is this relevant here? It does not relate to the subject of this sentence, which is describing what the stripping coil contained. Revise and make this its own sentence with a clear subject.

The sentence has been removed as suggested.

Line 150: 'particulate matter filter' - Not clearly labeled in Figure 2. Requires further description. Please add.

Further description has been added in Figure 1 as suggested.

[Figure]

Fig. 1. Schematic diagram of SAC-LOPAP. (M: gas flowmeter; S.S.: stripping solution; STA.: standard solution; D.S.: derivatization solution; Sec. Bottle: security bottle; C.B.T: cooling buffer tube; T.V.: triple valve for switching stripping solution and standard solution; P1, P2, P3, P4, P5: peristaltic pump for transferring solutions; FT: syringe filter; IC: integrated circuit. The gas flow

rate can be controlled from 0.2-2.0 L min$^{-1}$, with an optimal gas flow rate of 0.7L min$^{-1}$. The liquid flow rate can be controlled from 0.1-1.0 ml min$^{-1}$, with an optimal stripping liquid flow rate of 0.49ml min$^{-1}$).

Lines 151-152: 'but a large deviation … experiment' - Maybe, but this needs to be shown factually. Adding regular assessment of the system drift through use of online sampling of zero air (or pure N2, or by turning off the gas sampling flow and analyzing the reagents only) would give a mechanism to assess this. This is fairly common in benchtop versions of this instrument, as well as in the HONO LOPAP via the second stripping coil.

In fact, we add regular assessment of the system drift through use of online sampling of pure N$_2$, the range of blank signal in continuous operation for 48h were 2.856 V ~ 2.848 V and 2.254 V ~ 1.834 V of reduced solution concentration and former high solution concentration, and the maximum offset were 0.3% and 18.6 %, respectively, the baseline of low concentration solution has better stability (Fig. 3).

[Figure]

Fig. 3. The blank time series of the NH$_3$ detector ran continuously for 48 h.(Low concentration: 0.75g L$^{-1}$ salicylic acid, 0.014 g L$^{-1}$ sodium nitroferricyanide, and 0.2 g L$^{-1}$ NaOH as R1, then the 0.188ml L$^{-1}$ Sodium hypochlorite and 1.5 g L$^{-1}$ NaOH as R2; High concentration: 1g L$^{-1}$ salicylic acid, 0.1 g L$^{-1}$ sodium nitroferricyanide, and 1 g L$^{-1}$ NaOH as R1, then the 0.5ml L$^{-1}$ Sodium hypochlorite and 3 g L$^{-1}$ NaOH as R2).

Line 152: 'the solution concentation' - What solution concentration? From your scrubbed gas sample? I cannot tell. Please clarify.

The "the solution concentation" has been revised as "stripping and chromogenic concentrations (solution concentration).

Line 154: 'as shown in Figure 3' - Is this figure showing absorbance that results from the formation of precipitate? I do not think it is; based on how this is evolving. This seems to be a plot for absorbance resulting from the analysis of a calibration standard in the system, for which the flow rates and concentrations of all the reagents are also critical to report.

It is also confusing why the Authors are reporting absorbance and the raw values of I and Io. What are these actually showing that enhances the discussion? Overall, this figure could be removed from the manuscript and simply replaced with this sentence: 'The maximum absorbance of an XX mM NH4+ standard solution was obtained at 18.75 mM OH-'.

This figure has been removed from the manuscript as suggested and replaced simply with this sentence: The maximum absorbance of a 100 $\mu g\, L^{-1}$ $NH_4^+$ standard solution was obtained at 18.75 mmol $L^{-1}$ $OH^-$.

Line 159: 'C6H4(OH)(COOH)' - Use the name of the reagent, not the chemical formula. All of these details belong in the methods section. The result of obtaining maximum absorbance of the target NH4+ at a minimum amount of OH- is pretty trivial optimization. It is also not convincing that the amount of NH4+ reacted in these experiments is relevant to the upper limit of what could be expected in a real atmospheric sample, which is important to demonstrate, as all values below this would also then be expected to react to completion.

All chemical formula in the paper has been revised as the name of the reagent: We used 0.75g $L^{-1}$ salicylic acid (TCI, 99.5%, Japan), 0.014 g $L^{-1}$ sodium nitroferricyanide (TCI, 99%, Japan), and 0.2 g $L^{-1}$ NaOH as stripping solution (R1). Then the 0.188ml $L^{-1}$ Sodium hypochlorite (Aladdin, active chlorine10%, China) and 1.5 g $L^{-1}$ NaOH as A derivatization reaction to produce the colored reaction product solution (R2).

Lines 163-164: Are percentage values appropriate here? Why not give the factor? That is: XX times lower.

The "96 %, 98 %, and 99 %" has been revised as "0.04, 0.02 and 0.006 times".

Lines 165-168: If you are not actually measuring this, you cannot write as if you know this is true. The amount of iron-containing precipitate 'should' be minimized based on the known chemistry of this solution system. The mechanism by which this exerts an effect on the baseline also needs to be clearly communicated (i.e. it increases absorbance by scattering or absorbing light, resulting in measurement bias). It is also very strange that the Authors have not implemented a routine acid cleaning procedure for the system to remove these precipitates. It would interrupt the sampling for a few minutes and also give a really large benefit to the baseline maintenance. This is so commonly used as to be part of SOPs for benchtop analyses of soil extracts of NH4+, for example.

In fact, we tried interrupting the sampling for a few minutes and implementing 5% hydrochloric acid for the system to remove these precipitates. However, the concentration changed greatly before and after each cleaning precipitation. In addition, once the precipitation was formed, it will take a long time to remove the precipitation, which will also increase the risk of contaminating the detector.

Line 172: None of the relevant details in Figure 4 are discussed. What is the source and concentration of the NH3 being sampled? Or what does high and low concentration refer to? Is this [OH-]? And why are the scales not the same on both axes? The current depiction seems to be skewing the observations. The Authors should normalize both traces so the relative drift from the initial conditions is presented, along with sufficient detail to communicate what is actually being presented in the figure. Last, they must discuss this appropriately in this section. Why is there no comparison to any literature in this section as part of the discussion?

Figure 3 has been corrected and the discussion of Figure 4 has been added to the article.

Importantly, adding regular assessment of the system drift through use of online sampling of pure $N_2$, the range of blank signal in continuous operation for 48h were 2.856 V ~ 2.848 V and 2.254 V ~ 1.834 V of reduced solution concentration and former high solution concentration, and the

maximum offset were 0.3% and 18.6%, respectively, the baseline of low concentration solution has better stability (Fig. 3).

[Figure]

Fig. 3. The blank time series of the NH$_3$ detector ran continuously for 48 h.(low concentration: 0.75g L$^{-1}$ salicylic acid, 0.014 g L$^{-1}$ sodium nitroferricyanide, and 0.2 g L$^{-1}$ NaOH as R1, then the 0.188ml L$^{-1}$ Sodium hypochlorite and 1.5 g L$^{-1}$ NaOH as R2; High concentration: 1g L$^{-1}$ salicylic acid, 0.1 g L$^{-1}$ sodium nitroferricyanide, and 1 g L$^{-1}$ NaOH as R1, then the 0.5ml L$^{-1}$ Sodium hypochlorite and 3 g L$^{-1}$ NaOH as R2).

Lines 174-176: The description here is again very hard to follow. The Authors need to make a substantial effort to define the different components of their system that require optimization, then apply that very specific terminology throughout, with the order communicated at: i) the end of the introduction, ii) in the methods, and iii) in the results and discussion. The conclusions can then synthesize the overall outcomes in the context of the literature.

Here, for example, 'content of components in the solution' could mean just about anything. Is this the target analyte, NH3? Is it specifically the [OH-] from the prior section? Is it the large reduction in all the derivatization reagent solution concentrations from the prior section?

Also, the authors need to decouple the measurement value (absorbance) from the chemical reaction details (i.e. extent of completion in converting NH3 to the final dye).

We have made corrections in the end of the introduction, methodology, results, and discussion as suggested. The content of components in the solution' is the large reduction in all the derivatization reagent solution concentrations from the prior section. However, we revised the "the reduction of the content of components in the solution will lead to a decrease in absorbance, so it is necessary to adjust the temperature to speed up the reaction process and achieve a higher absorbance" as "High temperature can accelerate the reaction process and achieve better measurement accuracy and precision".

Line 179: 'stability and detection range' - What does this mean? Why are measurements with respect to metrics for these two terms not presented?

The "Considering the stability and detection range of the instrument" has been revised as "Considering the continuous delivery of solutions (the stability of pipeline connections) and the detection limit (40.5 ppt)" as suggested.

Line 180-181: 'to detect low concentrations of ammonia gas' - This is an arbitrary statement. Was a determination of the LOD made? Seems unlikely since the collection fraction has not been ascertained.

We agree with the referee and we deleted this word "low".

Line 184: 'measurement range was background contamination up to…' - This does not make sense. Another example of careless writing. Revise for correctness, using appropriate analytical chemistry and instrumentation terminology.

The "indicating that the measurement range was background contamination up to 100 μg L$^{-1}$ for NH$_4^+$ solutions" has been revised as "Therefore, the approximate mixing ratio of NH$_3$ corresponding to the standard liquid concentration is 0-99.1 ppb".

Line 186: First time these flows are shown without any justification/optimization shown. When/how was the collected fraction of NH3 determined? The Authors stated this was done in the lab, but it hasn't been shown yet.

This is also an inappropriate way to first present instrument detection limits. All that has been assessed, is the reaction system and LWCC detection limits and the approach is not clearly communicated. What measurement was used? How was the average signal determined? How was the noise determined? What value of signal to noise was then calculated to represent the LOD? How many replicate observations were used, at what time resolution, and did this LOD stay constant over time (in particular, was this calculated before and after the field observations, to identify bias)?

Section 3.1 has been added to explain how to determine the collected fraction of $NH_3$

**3.1 Sampling efficiency**

$NH_3$ Standard gas of 54.96 ppb was used as the sample to be collected through two identical serial stripping coils, and the concentration of liquid samples collected by the two stripping coils was measured to calculate the capture efficiency. The calculation formula is as below.

$$\gamma_1 = \frac{c_1}{c_1+c_2} \times 100\% \tag{6}$$

Where, $\gamma_1$ denotes the collection efficiency of the first stripping coil, $c_1$ and $c_2$ denote the concentration of $NH_4^+$ trapped in the first stripping coil and the second stripping coil, respectively.

The collection efficiency of $NH_3$ from the R1 reached more than 99% under different $c_{NaOH}$, $F_l$, and $F_g$. Figure 2a and Figure 2b show that the $F_l$ and the $F_g$ had almost no influence on collection efficiency. Figure 2c shows that $c_{NaOH}$ of 1.25 mmol $L^{-1}$ achieved the greatest collection efficiency in the R1 (99.9%). Therefore, the $c_{NaOH}$ of 1.25 mmol $L^{-1}$ was selected as the R1 of the $NH_3$. And we selected $F_l$ as 0.49 ml /min and $F_g$ as 0.7 L $min^{-1}$ in order to achieve the required detection range in this study.

[Figure]

Fig. 2. The absorption efficiency of stripping coil versus (a) gas flow rate ($c_{NaOH}$ = 4.0 mmol $L^{-1}$, $F_l$ = 0.49 ml $min^{-1}$), (b) liquid flow rate ($c_{NaOH}$ = 4.0 mmol $L^{-1}$, $F_g$ = 0.7 L $min^{-1}$), (c) concentration of NaOH in R1 ($F_l$ = 0.49 ml $min^{-1}$, $F_g$ = 0.7L $min^{-1}$).

Line 189: 'reducing the temperature' - Temperature of what?

It means lowering the temperature of the reaction coil.

Line 194: There are two points outside of the linear fit in panel b. No discussion about why these are outside the limit of linearity for the instrument. Is this poor performance of the detector? Incomplete reaction of NH4+ to the dye product? Something else?

However, due to the incomplete reaction of $NH_4^+$ with dye products, there are two points outside of the linear fit. Therefore, the approximate mixing ratio of $NH_3$ corresponding to the standard liquid concentration is 0-99.1 ppb, which is more than adequate for monitoring urban areas.

Line 195: Same issue with this figure as with Figure 3. There is no utility in presenting the measured I and I0 voltages.

We deleted Figure 5(a) as suggested.

The concentration of NH4+ sampled, and how it was delivered to the system, is not described/justified.

We have supplemented this part as suggested: R1 was used as diluent and the concentrations were 10, 25, 50, 75, 100, 150, and 200 μg L$^{-1}$ of $NH_4^+$ standard solution. In the calibration process, high purity $N_2$ was used as blank gas into the sampling tube, and the standard solution entered the solution system instead of the R1, then the measured absorbance values were used as absorbance-standard solution concentration plot and regression calculation.

The caption does not provide sufficient detail to understand what is being presented. For example, 'concentration' in panel b could be for anything. Is this supposed to be for $NH_4^+$?

The "concentration" in panel b was for $NH_4^+$, and we have corrected the figure as suggested.

Line 196: This section should be presented before 3.2.

We introduced this section before Section 3.2 as suggested.

Line 198: What is 'chrominance'? Do you mean molecular absorption spectrophotometry?

Yes, "chrominance" refers to molecular absorption spectrophotometry. We modified the text accordingly.

Line 198: 'residence time' - Why is there no presentation of the reaction rates that underpin the required residence time in the reaction coil? No stoichiometry or flow rate optimization is presented. Even if this is directly copied from an established standard operating procedure, this should be noted, along with the concentrations and flows of all the reagents.

Bianchi said that the signal delay caused by the reaction time reached 540s when the reagent of higher concentration was used. According to our previous experiment, this color reaction could not be completely reacted within a relatively short time (within 10 minutes). After the reagent was drained from the reaction pool, it was still carried out in the back-end connecting pipeline. It is not necessary to recognize this problem from the data of reaction rate. Therefore, we can do conditional tests to optimize the reaction conditions.

Line 202: 'new type of photoelectric detection to bubbly flow' - Why is this not presented in more detail? The writing here can also be substantially clarified.

The "new type of photoelectric detection" has been revised as "A commercialized liquid flow meter (SLI-1000, Sensirion, Switzerland)".

Line 205: 'the bump up of' - Should be 'increase'.

We agree with the referee and the "the bump up of" has been revised as "increase".

Line 209: 'chromogenic' – Just call it a derivatization reaction to produce the colored reaction product.

We agree with the referee and the "chromogenic" has been revised as "a derivatization reaction to produce the colored reaction product".

Line 217: 'consecutive measurements' - This is an inappropriate assessment of stability. How long did these four replicates require for collection? Shouldn't these be done across at least 4 days of constant instrument operation? Or maybe even 4 weeks? If it is going to be applied to real atmospheric sampling, it is not convincing that the performance is going to be as good as these values.

The "consecutive measurements" has been revised as "consecutive measurements (the collection of the four replicates were completed during a 4-week of constant instrument operation)".

Line 219: 'showing good stability' - No information on how the solutions were made, frequency of replacement, or sealed from absorbing ambient NH3. The details of the blank test also need to be presented. Was the inlet of the scrubbing coil overflown with clean N2 or zero air? Or was the gas sampling stopped, and a reagent blank measurement conducted?

We made the following additions to the article according to the suggestions: R1 was used as diluent and the concentrations were 10, 25, 50, 75, 100, 150, and 200μg $L^{-1}$ of $NH_4^+$ standard solution. In the calibration process, high purity $N_2$ was used as blank gas into the sampling tube, and the standard solution entered the solution system instead of the R1. Fig. 4 showed the calibration with the $NH_4^+$ concentration gradient of 0, 10, 25, 50 and 100 μg $L^{-1}$ (150, and 200μg $L^{-1}$ of ammonium ion standard solution were out of the detection range, which was discussed in section 3.4). Each concentration point was run for 40 minutes, and the RSD calculated from four consecutive measurements (the collection of the four replicates were completed during a 4-week of constant instrument operation) ranged from 0.32 % to 2.65 %, with the k varying from 0.0037 to 0.0040. Moreover, the blank experiment tests were automatically made every one or two days, that is, high purity $N_2$ was used as a blank gas through the sample tube for 40 minutes, the RSD of the blank signal in continuous operation for one month was 1.8 %, which indicated good repeatability and stability of the instrument.

[Figure]

Fig. 4. Calibration curves of standard solution with the same concentration gradient 4 times

Table 1. Linear regression with the same concentration gradient 4 times

| Time | k | b | $R^2$ |
|------|------|------|--------|
| 1st | 0.0037 | 0.0018 | 0.9998 |
| 2nd | 0.0039 | 0.0046 | 0.9996 |
| 3rd | 0.0040 | 0.0034 | 0.9997 |
| 4th | 0.0040 | 0.0003 | 0.9999 |

Line 220: 'was approximately 140 s' - This is actually an important observation to present the measurements from. Is this response time independent of NH4+ concentration?

In fact, "140 s" refers to the delay time, we have revised this sentence as follows: Seven switching samples were performed with 50 μg L$^{-1}$ NH$_4^+$ standard solution and R1, after calculating 10-90 % of the full signal after a change in concentration, the time response was approximately 140 s, which was much quicker than the method described by Bianchi et.al (measured to be 10 min) (Bianchi et al., 2012).

Line 221: 'by Bianchi' - Inappropriate way to do this citation. Revise.

The "by Bianchi" has been revised as "by Bianchi et.al".

Line 223: In Figure 6, why is a legend for each replicate appropriate, as opposed to the equation and regression coefficient for each calibration? Why is the y-axis label 'concentration' instead of '[NH4+]'? How long was each point measured for? Are there error bars that can be placed on each calibration point? Are these take into account in the regression? The lines on this figure also seem to be connecting the data points instead of representing the linear regressions, which is not useful. Revise.

We agree with the referee and the figure has been corrected.

[Figure]

Table 1. Linear regression with the same concentration gradient 4 times

| Time | k | b | $R^2$ |
|------|------|------|------|
| 1st | 0.0037 | 0.0018 | 0.9998 |
| 2nd | 0.0039 | 0.0046 | 0.9996 |
| 3rd | 0.0040 | 0.0034 | 0.9997 |
| 4th | 0.0040 | 0.0003 | 0.9999 |

Fig. 4. Calibration curves of standard solution with the same concentration gradient 4 times

Line 224: This table is not even discussed in the preceding section. Why bother presenting it? These 'number' values should be converted to values for the duration in time during continuous sampling.... If this is all within a few minutes to hours, it is not particularly appropriate. Overall, the scientific quality of this table is very poor.

The table has been deleted as suggested.

Lines 226-241: This all belongs in the methods section, just like any other of the hundreds of instrument intercomparisons that have been published previously. Lots of details are missing: What was the total combined sampling flow pulled through the inlet? And what was the resulting residence time of a gas sample in this inlet? Was the field container climate controlled? To what average values (and variability)?

 This was put into the methods section as suggested.

The inter-comparison experiment was conducted at the College of Environment Sciences and Engineering, Peking University, located within the 4th ring road in northern Beijing, China (39.59° N, 116.18°E). A commercial instrument Picarro G2103 analyzer (Picarro, US) used for atmospheric $NH_3$ measurement based on the CRDS method was deployed concurrently with SAC-LOPAP in the comparison, which could be used to validate other instruments (Twigg et al., 2022). The experiment took place from 15 September 2021 to 15 October 2021, with the instruments installed in a field container. Two instruments shared an inlet and were deployed 2.5 m above the ground. A Polytetrafluoroethylene (PTFE) filter (25 μm thickness, 46.2 mm diameter, 2 μm pore size, Whatman, USA) is used in the front of the sample module to remove ambient aerosols, which is placed into a round filter holder made of perfluoro alkoxy (PFA). We changed the filter every day with the aim of avoiding uncertainties. After the filtration of the aerosols, the sample gas flow is delivered into a 3.8 m long 1/4 in Teflon tube (Entegris, I.D.= 4.35 mm), and a temperature-controlled metal heating wire (set at 35 °C ±0.1 °C) is wrapped around the sample tube and covered with thermo-isolation materials. We ran our instrument with an additional drag flow of 1.75 L min$^{-1}$ with aim to ensure the ambient residence time was about 7.8 msec for all instruments. Data acquisition times were different for the above instruments during the inter-comparison. The base reporting periods for Picarro and SAC-LOPAP were 1 s and 30 s. For the purposes of comparison, data from the two instruments presented in this section were averaged to 30 s. In addition, high purity $N_2$ as zero gas was injected into the sampling tube at a flow rate of 0.7 L min$^{-1}$ and carried out every 7 days at the start of the campaign. The standard air source comes from China Sichuan Zhongce Biaowu Technology Co., LTD. The quality management system of the company conforms to the recognized standard in the Chinese industry (GB/T9001-2016/ISO 9001:2015). The

composition was ammonia (5.08ppm) and nitrogen with the uncertainty was 2%. In the test, pure $N_2$ was used as the dilution gas to obtain the required concentration of ammonia standard gas. Calibrations were performed using combinations of concentrations at 1.32, 4.95, 9.59, 17.90 and 54.96 ppb from the cylinder. In addition, 4.95 ppb and 54.96 ppb standard gas were injected into the sample tube every 7 days after zaro point. The field container was controlled at 25 °C ±1 °C to reduce the impact of temperature fluctuations on measurement results.

Line 227: Coordinates not specific enough. Please revise. Should also be located after China at the end of the sentence.

The "(40° N, 116° E)" has been revised as "located within the 4th ring road in northern Beijing, China (39.59° N, 116.18°E)".

Line 233: Incorrect notation for inches. Suggest using 'in.' or ' " '. Also, Teflon is a trademarked name. The polymer used in most tubing is perfluoro alkoxy (PFA). Revise.

The "1/4″" has been revised as "1/4 in".

Line 234: 'with thermo-isolation materials' - What does this mean? Was it temperature controlled? Revise.

We revised the "with thermo-isolation materials" as "and a temperature-controlled metal heating wire (set at 35 °C ±0.1 °C) is wrapped around the sample tube and covered with thermo-isolation materials".

Line 234: 'Teflon' – and these are usually made of PTFE. What is the pore size? Diameter of the filter? What kind of filter holder was it put in to?

We agree with the referee and the sentence has been updated as suggested: A Polytetrafluoroethylene (PTFE) filter (25 µm thickness, 46.2 mm diameter, 2 µm pore size, Whatman, USA) is used in the front of the sample module to remove ambient aerosols, which is placed into a round filter holder made of perfluoro alkoxy (PFA). We changed the filter every day with the aim of avoiding uncertainties.

Line 238: 'averaged to 5 min' - Why? Shouldn't this be done at the shortest timescale possible (i.e. 30 s)? Please provide a justification.

We agree with the referee and the "averaged to 5 min" has been revised "30 s".

Line 239: 'zero point' – This should be done at the start and end of the campaign as well. Was it? Details of how this was done are missing/insufficient. Please provide flows, gas composition, etc.

In addition, High purity nitrogen as zero gas was injected into the sampling tube and carried out every 7 days at the start and end of the campaign as well.

Line 239: 'the standard gas' - Vague and inappropriate level of detail for publication. Revise. What is the certified value? Is this $NH_3$ in a cylinder or from a permeation device or something else? How was the certified value of the standard ascertained? How was the composition controlled/delivered? Standard addition to ambient air sample? Or provided in clean gas?

We rewrote the passage as suggested: Calibrations were performed using combinations of concentrations at 1.32, 4.95, 9.59, 17.90 and 54.96 ppb from the cylinder mixture. The standard air source comes from China Sichuan Zhongce Biaowu Technology Co., LTD. The quality management system of the company conforms to the recognized standard in the Chinese industry (GB/T9001-2016/ISO 9001:2015). The composition was ammonia (5.08 ppm) and nitrogen with the uncertainty is 2%. In the test, pure $N_2$ was used as the dilution gas to obtain the required concentration of ammonia standard gas.

Lines 240-241: 'so that they… ensured quality control' - This does not make sense. How does measuring 'zero gas' maintain stability? Do you need to flush the entire sampling inlet for 40 minutes with this to get a reasonable measurement of zero? If yes, does this mean that these observations are subject to substantial inlet effects? Please provide a plot showing these time responses using the field campaign inlet at the highest time resolution possible for both instruments. This should be compared and discussed in context of the extensive literature on NH3 inlet effects and their mitigation.

We agree with the referee and this sentence does not make sense in our data analysis. We deleted this sentence.

Line 247: 'which might be… both instruments' - Blank measurements should be used to correct the observational dataset PRIOR to comparison. Why is the regression analysis not shown? Did the regression take into account the uncertainty in the measurements from each instrument (or at least the variability from averaging each from their smallest time interval to 5 minutes)? There will be a substantial y-intercept here and the slope between the two measurements is also critical to report. Revise and make sure to properly discuss those results in the context of the substantial number of other NH3 measurement intercomparisons that have been conducted previously. No context provided here at all. Not even a discussion on the range making sense, diurnal patterns, etc.

We rewrote the experimental results according to the advice. The corresponding text is revised as follows:

In this study, the concentration of our instrument ranged from 1.32 ppb to 47.86 ppb with an average of $12.64 \pm 8.63$ ppb, which was close to the concentrations of Picarro ($12.76 \pm 8.57$ ppb). The response speed was similar, indicated that SAC-LOPAP responded in time to rapid changed in $NH_3$ concentration. The diurnal variation results showed that the concentrations measured by the two instruments were very similar, with our instrument slightly lower than Picarro by 0.72ppb (Fig. 6b). Furthermore, relatively good correlations for the $NH_3$ data observed by these instruments were achieved over a large dynamic range of concentration with a slope of 1.00 and an $R^2$ of 0.96 (Fig. 6c). And we found that most of the time there were good correlations between the two instruments within one day except for the data of 23th and 30th September. The regression slope for all the days with higher and lower slopes are 1.46 and 0.72, respectively. We performed in-situ testing of both systems with cylinder, we produced $NH_3$ concentrations of about 1.32, 4.95, 9.59, 17.90 and 54.96 ppb. Fig. 6d showed regression analyses of the $NH_3$ standard gas concentrations obtained with the two instruments. The $NH_3$ concentrations measured by picarro and our instrument were strongly correlated, with a slope of 1.01 and an $R^2$ of 0.99.

In general, our instrument run relatively stable with the standard deviation of zero gas during the one month of observations being within 26 ppt (Picarro: 23 ppt), which was far below our

detection limit. Furthermore, the drift of SAC-LOPAP and Picarro at 4.95 ppb were 3.5% and 2.8%, while the drifts of 54.96 ppb were 1.5% and 0.7%, which meant that our instrument could keep steady for a long time and it could be used for the continuous online measurement of low concentration of ambient air. More detailed inter-comparison for these NH₃ instruments will be analyzed in a future publication.

[Figure]

Fig. 6. (a) Time series of NH₃ concentration during the comparison, (b) Diurnal variation of NH₃ concentrations observed by Picarro and SAC-LOPAP, (c) Regression analysis of the NH₃ concentrations observed by Picarro and SAC-LOPAP, and (d) Regression analysis of different concentrations of Picarro and SAC-LOPAP NH3 standard gases.

Line 249: 'concentration at the five-minute resolution' – It would be better to see this at the 30 s timescale, as it would provide a more appropriate assessment of instrument response time capabilities.

We agree with the referee and the "concentration at the five-minute resolution" has been revised "concentration at the 30 s timescale".

Line 251: 'study could measure… accurately' - I disagree. Should you not be delivering calibration gas to both instruments at multiple concentrations to show that they compare well? No slope and no interpretation of the intercept from the regression analysis makes this conclusion fatally flawed

because it is an arbitrary declaration. This needs to be substantiated through a careful analysis of this dataset.

We agree with the referee and make the following additions to the article: Furthermore, relatively good correlations for the $NH_3$ data observed by these instruments were achieved over a large dynamic range of concentration with a slope of 1.00 and an $R^2$ of 0.96 (Fig. 6c). And we found that most of the time there were good correlations between the two instruments within one day except for the data of 23th and 30th September. The regression slope for all the days with higher and lower slopes are 1.46 and 0.72, respectively. We performed in-situ testing of both systems with cylinder, we produced $NH_3$ concentrations of about 1.32, 4.95, 9.59, 17.90 and 54.96 ppb. Fig. 6d showed regression analyses of the $NH_3$ standard gas concentrations obtained with the two instruments. The $NH_3$ concentrations measured by picarro and our instrument were strongly correlated, with a slope of 1.01 and an $R^2$ of 0.99.

Line 252: 'STD' - What is this? Standard deviation? Is this within any single sampling of zero gas? Or has the calculation been performed across all of the replicates? How much drift was there in the SAC over time? No discussion on this, despite all the prior sections saying that the tuned parameters are necessary to prevent drift.

Yes, "STD" refers to the Standard deviation. In general, high purity $N_2$ as zero gas was injected into the sampling tube and carried out every 7 days (section 2.4), our instrument run relatively stable with the standard deviation of zero gas during the one month of observations being within 26 ppt (picaro: 23 ppt), which was far below our detection limit.

Line 253: 'RSD of the standard gas within 0.76% (Table 2)' - Table 2 can be replaced with a single sentence for each of the experiments reported. One for the blank comparisons (no discussion of the difference in values between the two instruments is a big issue here). The second sentence would describe the conditions of the calibrations.

We agree with the referee and make the following update to the article: In general, our instrument run relatively stable with the standard deviation of zero gas during the one month of observations being within 26 ppt (picaro: 23 ppt), which was far below our detection limit. Furthermore, the drift

of SAC-LOPAP and Picarro at 4.95 ppb were 3.5% and 2.8%, while the drifts of 54.96 ppb were 1.5% and 0.7%, which meant that our instrument could keep steady for a long time and it could be used for the continuous online measurement of low concentration of ambient air. More detailed inter-comparison for these $NH_3$ instruments will be analyzed in a future publication.

Last, doing the calibrations at 40 ppb of $NH_3$ is not a fair assessment of 'agreement'. This should have been done at 5, 10, 20, and 40 ppb. Instead the Authors choose the highest observation from the ambient dataset, which is a concentration where agreement between the techniques would be most easy to obtain. Lower ambient values (100 pptv to 1 ppbv) are widely reported throughout the literature to present more of an issue, but this is neither presented, nor discussed. Major revision with respect to this is required.

We agree with the referee and 4.95 ppb and 54.96 ppb standard gas were injected into the sample tube every 7 days after zaro point. The results are as follows: Furthermore, the drift of SAC-LOPAP and Picarro at 4.95 ppb were 3.5% and 2.8%, while the drifts of 54.96 ppb were 1.5% and 0.7%, which meant that our instrument could keep steady for a long time and it could be used for the continuous online measurement of low concentration of ambient air.

Line 257: These are mixing ratios, not concentrations. The point of presenting panels b and c is not clear. They are not discussed at all. Overall, these three panels are insufficient analyses of the intercompared data for publication in a peer-reviewed article. Review the literature and revise by building on prior reports. Place this work into context of those prior studies as well.

We agree with the referee and make the following additions to the article: The time series of the concentration of $NH_3$ during the inter-comparison period of Picarro and SAC-LOPAP were presented in Fig. 6a. There were a few data gaps for the above instruments caused by calibration operations and instrument maintenance. Instruments display similar temporal features for $NH_3$ concentrations over the duration of the study. In this study, the concentration of our instrument ranged from 1.32 ppb to 47.86 ppb with an average of 12.64 ± 8.63 ppb, which was close to the concentrations of Picarro (12.76 ± 8.57 ppb). The response speed was similar, indicated that SAC-LOPAP responded in time to rapid changed in $NH_3$ concentration. The diurnal variation results

showed that the concentrations measured by the two instruments were very similar, with our instrument slightly lower than picarro by 1ppb (Fig. 6b). Furthermore, relatively good correlations for the $NH_3$ data observed by these instruments were achieved over a large dynamic range of concentration with a slope of 1.00 and an $R^2$ of 0.96 (Fig. 6c). And we found that most of the time there were good correlations between the two instruments within one day except for the data of 23th and 30th September. The regression slope for all the days with higher and lower slopes are 1.46 and 0.72, respectively. We performed in-situ testing of both systems with cylinder, we produced $NH_3$ concentrations of about 1.32, 4.95, 9.59, 17.90 and 54.96 ppb. Fig. 6d showed regression analyses of the $NH_3$ standard gas concentrations obtained with the two instruments. The $NH_3$ concentrations measured by picarro and our instrument were strongly correlated, with a slope of 1.01 and an $R^2$ of 0.99.

[Figure]

Fig. 6. (a) Time series of $NH_3$ concentration during the comparison, (b) Diurnal variation of $NH_3$ concentrations observed by Picarro and SAC-LOPAP, (c) Regression analysis of the $NH_3$ concentrations observed by Picarro and SAC-LOPAP, and (d) Regression analysis of different concentrations of Picarro and SAC-LOPAP NH3 standard gases.

There are five blank and calibration tests noted, but I only see two gaps in the reported dataset. Add markers to the full timeseries to indicate where each of these were performed. If they were not spaced out by actual ambient observations, then there is no value in some of these replicates.

The dotted line is when we did the blank experiment:

[Figure]

Line 260: Revise the conclusions section in its entirety based on all other changes required throughout the manuscript. This basically needs to be completely re-written since the contents of the manuscript are currently totally unacceptable for publication.

Ammonia ($NH_3$) in the atmosphere affects the environment and human health and is therefore increasingly recognized by policy makers as an important air pollutant that needs to be mitigated. The accurate and precise detection of ambient $NH_3$ concentrations is therefore an urgent need for the exploration of secondary pollution at the regional scale in China.

At the present stage, ambient $NH_3$ measurements at many supersites are still done with spectroscopic, mass spectrometric and wet chemical methods, which are restricted by the high detection limit and lower time resolution. In this study, we provide an online $NH_3$ monitoring system based on wet chemistry stripping and long path absorption photometer of atmospheric $NH_3$, our new SAC-LOPAP system has several significant improvements: one is the optimization of reaction conditions. The low concentration but higher flow rate of solutions decreases the precipitate's production, and the cooling buffer tube and the filter trap most of the precipitates. The others are the constant temperature module and liquid flow controller. The constant temperature module in the system reduces the influence of ambient temperature on the reaction process and color degree. Similarly, adding a liquid flow controller is helpful to the stability of the flow rate and further increases the stability of the reaction process. These improvements reduce the system error and significantly increase the sustainability of SAC-LOPAP operation. Our instrument reached a detection limit of about 40.5 ppt with a stripping liquid flow rate of 0.49 ml min$^{-1}$ and a gas sample flow rate of 0.70 L min$^{-1}$ in the current condition, and the measuring range of the instrument is 0-99.1 ppb. Our system has also been characterized in a laboratory setting where we can measure low concentrations. SAC-LOPAP and Picarro were compared in urban areas for a month with relatively

good agreement ($R^2$ = 0.967). In addition, the diurnal variation results showed that the concentrations of the two instruments were very similar, Therefore, we conclude that our update of the ammonia measurement experimental framework has been successful. However, more research about field measurement and comparison is needed to verify the equipment's performance in routine observation, and the influence of particulate ammonium on the results of $NH_3$ detection also requires further study.

---

## Author Comment (AC2)

Response to Referees

Manuscript Number: amt-2023-33

Manuscript Title: Improvement of online monitoring technology based on the Berthelot reaction and long path absorption photometer for the measurement of ambient NH$_3$: Field applications in low-concentration environments

The discussion below includes the complete text from the referees, along with our responses to the specific comments and the corresponding changes made to the revised manuscript.

The detailed answers to the individual referee's comments in blue.

All of the line numbers refer to the original manuscript.

Response to Referee #2 Comments:
* * *
We would like to thank the referee for his/her detailed comments and suggestions which helped us a lot to improve the quality of the paper. Our revised manuscript has been further edited by professional language services.

Review of "Improvement of online monitoring technology based on the Berthelot reaction and long path absorption photometer for the measurement of ambient NH$_3$: Field applications in low-concentration environments", by Tian (2023)

The authors report the development of a new technique to measure atmospheric ammonia (NH$_3$) using wet chemistry based on the Berthelot reaction and a long path absorption photometer (SAC-LOPAP). The manuscript provides detail on the optimization of reaction conditions within the sampling and reacting modules, as well as a 1-month field evaluation with a co-located commercial Cavity Ring Down Spectroscopy instrument (CRDS). Given challenges inherent to measuring NH 3, the authors should be commended for developing a new NH$_3$ measurement technique. However, the manuscript needs major revisions to refocus the paper so that the introduction and conclusions

are aligned with the results. Furthermore, additional information is needed in some places of the manuscript. General and specific comments are below.

General Comments:

The title and parts of the Introduction mention the low detection limit of the SAC-LOPAP and its ability to measure $NH_3$ in low-concentration environments. However, no detail is given on how the limit of detection (40.5 ppt) was quantified. Since the manuscript discusses how the low detection limit is one of the advantages of the SAC-LOPAP, there needs to be a detailed description of how the detection limit was calculated.

The field application of the SAC-LOPAP was not performed in a low $NH_3$ environment, with mixing ratios ranging from ~2 ppb to ~45 ppb. The title should be revised to clarify the field application was in an urban environment (i.e., Beijing) and not in a low $NH_3$ environment. Furthermore, the discussion throughout the manuscript should be reframed so that it focuses on the good agreement with the CRDS. Currently, portions of the manuscript focus on the SAC-LOPAP's ability to measure low concentrations and its low detection limit, neither of which have been shown or adequately explained.

We thank the reviewer for taking the time to review this study and providing the constructive feedback of the manuscript.

Specific Comments:

Line 20 – state what the "established system" is in the abstract.

The "established system" has been revised as "a commercial instrument Picarro G2103 analyzer (Picarro, US)".

Line 21 – state what the "good correlation" is, as well as other relevant statistical metrics.

The "good correlation" has been revised as "good correlation with a slope of 1.00 and an $R^2$ of 0.96"

Line 38 – this should read "sulphuric acid" instead of "sulfate"; also, please define CLOUD.

We think this sentence is inappropriate. we deleted it.

Lines 42-43 – clarify that this study is specific to China, and please verify that the citation is correct.

We think this sentence is inappropriate. we deleted it.

Line 62 – define DFB.

We think this statement is inappropriate. We deleted it.

Line 66 – what is meant by "NH$_3$ species"? This should be "NH$_3$", "NHx species", or "NRx species".

We agree with the referee and the "NH$_3$ species" has been revised as "NH$_3$".

Lines 67-68 – what does "special materials" mean?

We think this statement is inappropriate. We deleted it.

Lines 69 and 73 – there appear to be in-text citations errors, please correct/verify.

We agree with the referee. we deleted this sentence.

Lines 75-76 – discuss the importance of inlet design on ambient measurements of NH$_3$ and detection limits.

We agree with the referee that the inlet is important for ambient measurements of NH$_3$ and detection limits. However, this article is mainly aimed at the discussion of new instruments. More detailed inter-comparison about inlet for these NH$_3$ instruments will be analyzed in a future publication.

Line 89 – clarify what "statically" means and quantify what a "long time" is (e.g., months, years?).

We agree with the referee and we added this sentence: that it can be set up and left to run for long periods of time stably for one months used for the continuous online measurement of low concentrations ammonia of ambient air.

Lines 92-93 – be quantitative with what is meant by "low-concentration" and "low detection limit".

We agree with the referee and the "Our instrument is designed to measure NH$_3$ in a low-concentration environment with the good stability, low detection limit and small size." has been revised as "Our instrument is designed to measure NH$_3$ in a low-concentration environment (under 20 ppb) with the good stability, low detection limit (less than 60 ppt) and small size".

Line 104 (Figure 1) – verify that reactants and products balance in the reaction scheme (e.g., the HCl appears to be missing from Step 2).

This equation does not list all the reaction products, only the major reactants and products.

Line 111 – clarify what "NH$_3$ components" means.

The "NH$_3$ components" has been revised as "NH$_3$".

Line 125 – correct "invert" to "convert" and provide the units for C NH$_4^+$.

The "invert" has been revised as "convert" and "Where $C_{NH_3}$ denotes the content of NH$_3$ in the air sample" has been revised as "Where $C_{NH_3}$ denotes the content of NH$_3$ in the air sample (ppb)".

Line 126 – what is meant by gas "production sample"?

The "NH₃ concentration in the gas production sample $C_{NH_3}$" has been revised as "NH₃ concentration in the gas $C_{NH_3}$".

Lines 129-130 – should the temperature be for ambient air since the equation was derived by the Ideal Gas Law?

In fact, the "temperature" refers to room temperature, because our gas flow is measured at room temperature.

Lines 131-132 – please provide detail on the capture efficiency parameter. How was it determined, what does it depend on, and how sensitive is it to the set-up (e.g., temperature of the solution, pH of the solution, inlet design)?

We agree with the referee and we added the text:

**3.1 Sampling efficiency**

NH₃ Standard gas of 54.96 ppb was used as the sample to be collected through two identical serial stripping coils, and the concentration of liquid samples collected by the two stripping coils was measured to calculate the capture efficiency. The calculation formula is as below.

$$\gamma_1 = \frac{c_1}{c_1 + c_2} \times 100\% \tag{6}$$

Where, $\gamma_1$ denotes the collection efficiency of the first stripping coil, $c_1$ and $c_2$ denote the concentration of NH₄⁺ trapped in the first stripping coil and the second stripping coil, respectively.

The collection efficiency of NH₃ from the R1 reached more than 99% under different $c_{NaOH}$, $F_l$, and $F_g$. Figure 2a and Figure 2b show that the $F_l$ and the $F_g$ had almost no influence on collection efficiency. Figure 2c shows that $c_{NaOH}$ of 1.25 mmol L⁻¹ achieved the greatest collection efficiency in the R1 (99.9%). Therefore, the $c_{NaOH}$ of 1.25 mmol L⁻¹ was selected as the R1 of the NH₃. And we selected $F_l$ as 0.49 ml /min and $F_g$ as 0.7 L min⁻¹ in order to achieve the required detection range in this study.

[Figure]

Fig. 2. The absorption efficiency of stripping coil versus (a) gas flow rate ($c_{NaOH}$ = 4.0 mmol L⁻¹, $F_l$

= 0.49 ml min$^{-1}$), (b) liquid flow rate ($c_{NaOH}$ = 4.0 mmol L$^{-1}$, $F_g$ = 0.7 L min$^{-1}$), (c) concentration of NaOH in R1 ($F_l$ = 0.49 ml min$^{-1}$, $F_g$ = 0.7L min$^{-1}$).

Line 143 – rather than say "and so on", list of all the measures needed to achieve continuous online measurement.

We agree with the referee and list of all the measures needed to achieve continuous online measurement. We deleted the "and so on".

Line 150 – should this be precipitate instead of sediment?

The "sediment" has been revised as "precipitate".

Line 156 – at a pH of ~12 and temperature of 55°C, gaseous ammonia won't be as soluble as under typical wet chemistry methods. What effect do the high pH and high temperature on the capture efficiency of ammonia?

Ammonia gas is captured by the stripping solution, which is the mixed solution of salicylic acid and sodium hydroxide with a pH of 5.0, which is weakly acidic. Under the flow condition in this paper, the stripping solution can capture more than 99% of ammonia gas. And cycling water kept the trap at room temperature, and did not reach 55□ in the reaction zone. The pH of ~12 and temperature of 55°C mentioned in this paper are in the reaction zone and will not affect the ammonia trapping

Line 172 (Figure 4) – Define "high" and "low" concentration.

We defined "high" and "low" concentration in Figure 4 as suggested.

[Figure]

Fig. 4. The blank time series of the NH$_3$ detector ran continuously for 48 h.(low concentration: 0.75g L$^{-1}$ salicylic acid, 0.014 g L$^{-1}$ sodium nitroferricyanide, and 0.2 g L$^{-1}$ NaOH as R1, then the 0.188ml L$^{-1}$ Sodium hypochlorite and 1.5 g L$^{-1}$ NaOH as R2; High concentration: 1g L$^{-1}$ salicylic acid, 0.1 g

L$^{-1}$ sodium nitroferricyanide, and 1 g L$^{-1}$ NaOH as R1, then the 0.5ml L$^{-1}$ Sodium hypochlorite and 3 g L$^{-1}$ NaOH as R2).

Lines 185-186 – Describe in detail how the detection limit was determined.

We describe in detail how detection limits are determined as suggested: "however, due to the incomplete reaction of NH$_4^+$ with dye products, there are two points outside of the linear fit (standard solution concentrations are 150 and 200 μg L$^{-1}$). Therefore, the approximate mixing ratio of NH$_3$ corresponding to the standard liquid concentration is 0-99.1 ppb. The detection limit for NH$_4^+$ liquid solution is about 40.9 ng L$^{-1}$, which is calculated as 3 times the average standard deviation of blank signal noise in one hour. With an air sample flow rate of 0.7 L min$^{-1}$ and a liquid flow rate of 0.49 ml min$^{-1}$, this translates to a gas phase mixing ratio of about 40.5 ppt".

Lines 190-191 – Describe in detail how the upper range was determined.

We added the text as suggested:

The standard solution entered the solution system instead of the absorption solution, then the measured absorbance values were used as absorbance-standard solution concentration plot and regression calculation (The experimental process is described in Section 3.3). The result is shown in Fig. 5, a high degree of correlation was found between the standard solution and absorbance with a correlation coefficient of R$^2$ = 0.99 for the standard solution of 0-100 μg L$^{-1}$, however, due to the incomplete reaction of NH$_4^+$ with dye products and the limitation of the detector, there are two points outside of the linear fit in panel b. Therefore, the approximate mixing ratio of NH$_3$ corresponding to the standard liquid concentration is 0-99.1 ppb.

Line 215 – at a pH ~12 most of the NHx would NH$_3$, and not NH$_4^+$.

The "NH$_4^+$" has been revised as "NH$_3$".

Line 222 (Figure 6) – describe what approximate mixing ratio of NH$_3$ the calibration concentrations correspond to.

We agree with the referee and we added the describe in Section 3.3.

"R1 was used as diluent and the concentrations were 10, 25, 50, 75, 100, 150, and 200μg L$^{-1}$ of NH$_4^+$ standard solution. In the calibration process, high purity N$_2$ was used as blank gas into the sampling tube, and the standard solution entered the solution system instead of the absorption solution. Fig. 5 showed the calibration with the NH$_4^+$ concentration gradient of 0, 10, 25, 50 and 100 μg L$^{-1}$ (150, and 200μg L$^{-1}$ of ammonium ion standard solution were out of the detection range,

which was discussed in section 3.4)".

Line 233 – did the CRDS and SAC-LOPAP share an inlet, or did each have its own inlet at the same height?

The CRDS and SAC-LOPAP share an inlet.

Lines 249-251 – recommend including other statistical metrics in addition to R2, such as mean bias or slope of linear regression.

We agree with the referee. The "The $NH_3$ concentrations measured by those instruments were strongly correlated ($R^2 = 0.967$), which significantly indicated that the SAC-LOPAP developed in this study could measure the $NH_3$ concentration accurately." has been revised as "relatively good correlations for the $NH_3$ data observed by these t instruments were achieved over a large dynamic range of concentration with a slope of 1.00 and an $R^2$ of 0.96".

Line 261 – most of the Conclusions section describes improvements made to the Berthelot reaction conditions, and do not necessarily represent an improved methodology for quantifying ambient $NH_3$ relative to other $NH_3$ measurement technologies. This sentence should be rephrased to clarify this nuance.

We agree with the referee and we also think this sentence is inappropriate. We deleted the sentence.

Lines 272-273 – as noted above, there was insufficient explanation for how the detection limit was determined.

We describe in detail how detection limits are determined as suggested: "however, due to the incomplete reaction of $NH_4^+$ with dye products, there are two points outside of the linear fit (standard solution concentrations are 150 and 200 μg $L^{-1}$). Therefore, the approximate mixing ratio of $NH_3$ corresponding to the standard liquid concentration is 0-99.1 ppb. The detection limit for $NH_4^+$ liquid solution is about 40.9 ng $L^{-1}$, which is calculated as 3 times the average standard deviation of blank signal noise in one hour. With an air sample flow rate of 0.7 L $min^{-1}$ and a liquid flow rate of 0.49 ml $min^{-1}$, this translates to a gas phase mixing ratio of about 40.5 ppt".

---

## Author Response (AR2)

Response to Referees

Manuscript Number: amt-2023-33

Manuscript Title: Colorimetric derivatization of ambient ammonia (NH$_3$) for detection by long path absorption photometry

The discussion below includes the complete text from the referee, along with our responses to the specific comments and the corresponding changes made to the revised manuscript.

The detailed answers to the individual referee's comments in blue.

All of the line numbers refer to the original manuscript.

Response to Referee Comments:

We would like to thank the referee for his/her detailed comments and suggestions which helped us a lot to improve the quality of the paper. Our revised manuscript has been further edited by professional language services.

The authors introduce an innovative and online technique for the precise wet chemical measurement of atmospheric ammonia (NH$_3$). This technique is based on the salicylic acid derivatization reaction in conjunction with a long path absorption photometer (SAC-LOPAP). Through refinement of reaction parameters, incorporation of a consistent temperature control module, and integration of a liquid flow controller, the online NH$_3$ monitoring system demonstrates prolonged operational stability. The study also provides the outcomes from a comprehensive 1-month field assessment, conducted alongside a commercially available Cavity Ring Down Spectroscopy instrument (CRDS), ensuring the reliability and accuracy of the developed approach. Given the persisting complexities in the measurement of environmental NH$_3$, the authors warrant commendation for their pioneering efforts in advancing online NH$_3$ measurement techniques.

Comments:

1. Line 27: The manuscript refers to "recent research", but the references provided are not from the last three years.

Many thanks to the reviewer for the comment. References from the last three years have been added to the article

2. Line 33: Pay attention to the subscript notation of $NH_3$.

The "NH3" has been revised as "$NH_3$" according to the journal guidelines.

3. Line 55: Is "traceNH3" intended to be denoted as "trace NH3"?

The "traceNH$_3$" has been revised "trace NH$_3$" as suggested.

4. Line 59: Could "The NH3 mining ratio" potentially be intended as "The NH3 mixing ratio"?

The "The NH$_3$ mining ratio" has been revised "The NH$_3$ mixing ratio" as suggested.

5. Line 75: What is "long-1 path"?

This sentence was miswritten. The "long-1 path" has been revised "long path".

6. Line 111: How to determine the temperature of circulating cooling water outside the stripping coil? Why set 10-15 ° C? What happens if the temperature is too high or too low? There is no clear explanation in the manuscript.

The sampling temperature could be kept low (10~15 °C) by using the cooling device, which favors the capture of gaseous pollutant. In addition, this temperature could satisfy 99% of the sampling efficiency according to Section 3.1.

7. Figure 1 shows the debubbler after the reaction coil, is there always a bubble in the pipeline before that? Does it affect the efficiency of the derivatization reaction? Why not add a debubbler after the stripping coil?

Yes, there was always a bubble in the pipeline before reaction coil and it did not affect the efficiency of the derivatization reaction. In this study, the Segmented Continuous Flow Analysis (SCFA) created by American professor Skeggs was used as reference, which could speed up the sample update speed and shorten the response time compared with the condition without bubbles in the pipeline. Furthermore, the purpose of added a debubbler was to prevent bubbles from entering the LWCC. If added a debubbler after the stripping coil, it was possible for bubbles to re-enter LWCC.

8. Line 162-164: "the sample gas……" Is there any adsorption loss of NH3 in Teflon tube? Have any experiments been carried out to substantiate this?

There was a small adsorption loss of $NH_3$ in Teflon tubes. The influence of temperature, humidity, correction methods on the adsorption loss of $NH_3$ in Teflon tube were carried out, in which humidity was the key factor, so it was very important to keep the sample tube insulated.

9. Figure 2a and 2b show that Fl and Fg have no effect on collection efficiency. Then why do the optimal Fl and Fg appear? What is the definition of optimal flow rate?

Yes, $F_l$ and $F_g$ had no effect on collection efficiency. Because $NH_3$ was "sticky" and highly soluble in water, it had little effect on collection efficiency. However, the $F_l$ and $F_g$ were parameters that affect the measurement range of $NH_3$ according to Formula 5. The peristaltic pump and air pump were relatively stable in the range of 0.3-0.7 ml min$^{-1}$ and 0.4-1.0 L min$^{-1}$ in this instrument, respectively, so the middle values were taken as the parameter.

10. Line 198: What does "temperatureto" refer to?

This sentence was miswritten. The "temperatureto" has been revised "temperature".

11. There is a problem with the color of the curve corresponding to the left and right ordinates in Figure 3.

We corrected the Figure 3 as suggested.

[Figure]

Fig. 3. The blank time series of the NH$_3$ detector ran continuously for 48 h.(Low concentration: 0.75g L$^{-1}$ salicylic acid, 0.014 g L$^{-1}$ sodium nitroferricyanide, and 0.2 g L$^{-1}$ NaOH as R1, then the 0.188ml L$^{-1}$ Sodium hypochlorite and 1.5 g L$^{-1}$ NaOH as R2; High concentration: 1g L$^{-1}$ salicylic acid, 0.1 g L$^{-1}$ sodium nitroferricyanide, and 1 g L$^{-1}$ NaOH as R1, then the 0.5ml L$^{-1}$ Sodium hypochlorite and 3 g L$^{-1}$ NaOH as R2).

12. Line 254: "RSD" is introduced here for the first instance; it is advisable to provide its complete expansion or offer an explanation.

We agree with the referee. We revised "RSD" as "relative standard deviation (RSD)".

13. In lines 287-289, it is highlighted that within the prevailing instrument settings, the discernible range for NH3 concentrations spans from 40.5 ppt to 99.1 ppb. However, the "Conclusion" section denotes the instrument's detection range as 0-99.1 ppb. Do these two references indeed pertain to the identical detection span, and if so, what accounts for the numerical disparity?

Yes, the discernible range for NH$_3$ concentrations spans from 40.5 ppt to 99.1 ppb under existing conditions. We made a mistake in the "Conclusion" section and revised "0-99.1 ppb" as "40.5 ppt to 99.1 ppb".

14. The manuscript indicates that two identical serial stripping coils are used in series, but it is not clear what the role of the second stripping coils is. Just to calculate the NH3 collection efficiency?

Or is it the same as the HONO-LOPAP to eliminate the effects of distractions? There is no detail in the manuscript.

Yes, two identical serial stripping coils were used in series to calculate the $NH_3$ collection efficiency. In fact, the role of the second stripping coil of HONO-LOPAP was to evaluate the interference of $NO_2$ in the air, while gaseous ammonia was very stable and there is no interference from other components in the atmosphere. Importantly, the first stripping coils of SAC-LOPAP has a very high sampling efficiency with 99%, so a second channel was not needed.

15. The manuscript lacks detailed information regarding the frequency of filter replacements preceding LWCC and the expected lifespan of R1 and R2 solutions. Could you provide further clarification on these aspects?

We agree with the referee and we added this sentence in section 2.3 "The ideal use cycle of R1 and R2 was half a month" and in section 3.3 "and the filter was changed weekly."

16. It is mentioned in the introduction that ambient NH3 is 5 ppt-500 ppb, but the instrument in the manuscript can measure 99.1ppb according to the existing conditions. Elaborating in detail on the specific conditions that necessitate modification, along with quantifying the potential expansion of the detection range, and supplementing these insights with an accompanying data chart, would substantially enhance the richness of the article's content.

 We agree with the referee and we added this sentence in section 3.4 "For example, the following table could be obtained according to Formula 5 and the stability ranges of $F_l$ and the $F_g$. The detection limit could be reduced to 14.47 ppt and the detection upper limit can be increased to 519.02 ppb by adjusting the $F_l$ and the $F_g$ (Table.2)".

Table 2. Relationship between $F_l$ , $F_g$ and detection range of SAC-LOPAP

| $F_l$, ml min$^{-1}$ | $F_g$, L min$^{-1}$ | C(NH$_3$)$_{min}$, ppt, [NH$_4^+$] = 40.9 ng L$^{-1}$ | C(NH$_3$)$_{max}$, ppb, [NH$_4^+$] =100 µg L$^{-1}$ |
|---|---|---|---|
| 0.25 | 1 | 14.47 | 35.38 |
| 0.35 | 0.85 | 23.84 | 58.29 |
| 0.5 | 0.7 | 41.35 | 101.11 |
| 0.75 | 0.4 | 108.55 | 265.41 |
| 1.1 | 0.3 | 212.28 | 519.02 |